# YAP and TAZ are transcriptional co-activators of AP-1 proteins and STAT3 during breast cellular transformation

Lizhi He[1†], Henry Pratt[2†], Mingshi Gao[2†], Fengxiang Wei[3], Zhiping Weng[2]*, Kevin Struhl[1]*

[1]Department of Biological Chemistry and Molecular Pharmacology, Harvard Medical School, Boston, United States; [2]Program in Bioinformatics and Integrative Biology, University of Massachusetts Medical School, Worcester, United States; [3]Genetics Laboratory, Shenzhen Longgang District Maternity and Child Healthcare Hospital, Shenzhen, China

*For correspondence:
zhipingweng@gmail.com (ZW);
kevin@hms.harvard.edu (KS)

[†]These authors contributed
equally to this work

**ABSTRACT** The YAP and TAZ paralogs are transcriptional co-activators recruited to target sites by TEAD proteins. Here, we show that YAP and TAZ are also recruited by JUNB (a member of the AP-1 family) and STAT3, key transcription factors that mediate an epigenetic switch linking inflammation to cellular transformation. YAP and TAZ directly interact with JUNB and STAT3 via a WW domain important for transformation, and they stimulate transcriptional activation by AP-1 proteins. JUNB, STAT3, and TEAD co-localize at virtually all YAP/TAZ target sites, yet many target sites only contain individual AP-1, TEAD, or STAT3 motifs. This observation and differences in relative crosslinking efficiencies of JUNB, TEAD, and STAT3 at YAP/TAZ target sites suggest that YAP/TAZ is recruited by different forms of an AP-1/STAT3/TEAD complex depending on the recruiting motif. The different classes of YAP/TAZ target sites are associated with largely non-overlapping genes with distinct functions. A small minority of target sites are YAP- or TAZ-specific, and they are associated with different sequence motifs and gene classes from shared YAP/TAZ target sites. Genes containing either the AP-1 or TEAD class of YAP/TAZ sites are associated with poor survival of breast cancer patients with the triple-negative form of the disease.

## Introduction

The Hippo signal transduction pathway plays critical roles in development, homeostasis, and tumor progression (*Piccolo et al., 2014*; *Varelas, 2014*; *Yu et al., 2015*; *Totaro et al., 2018*; *Moya and Halder, 2019*). Internal and external signals relayed through the Hippo pathway converge on YAP and TAZ, paralogous proteins with 63 % sequence similarity that are the major effectors of this pathway (*Piccolo et al., 2014*; *Totaro et al., 2018*; *Ma et al., 2019*). Once activated, YAP/TAZ translocate into nucleus and act as transcriptional co-activators, most notably by interacting with the TEAD family of DNA-binding transcription factors (*Piccolo et al., 2014*; *Totaro et al., 2018*; *Ma et al., 2019*; *Moya and Halder, 2019*).

Deregulation of YAP/TAZ activity is frequently observed in various human cancers, contributing to cancer initiation, progression, and metastasis (*Johnson and Halder, 2014*; *Yu et al., 2015*; *Zanconato et al., 2016*). YAP/TAZ activation is also linked to chemo-resistance in cancer therapy (*Johnson and Halder, 2014*; *Yu et al., 2015*; *Zanconato et al., 2016*). Genetic experiments indicate that YAP and TAZ have distinct and overlapping cellular functions (*Plouffe et al., 2018*; *Shreberk-Shaked et al., 2020*), but the molecular basis for the distinct functions is unknown. More generally, the transcriptional

mechanisms and regulatory circuits mediated by YAP/TAZ in cancers beyond those associated with TEAD proteins are not well understood.

In addition to directly interacting with the TEAD family of DNA-binding proteins, YAP/TAZ can function through other mechanisms. Via its interaction with TEAD proteins, YAP/TAZ can synergize with JUN/FOS at composite regulatory elements containing both TEAD and AP-1 motifs (*Zanconato et al., 2015*; *Liu et al., 2016*). YAP-TEAD signaling cooperates with AP-1 to promote basal cell carcinoma (*Maglic et al., 2018*) and pancreatic cancer in mice (*Park et al., 2020*), and YAP/TAZ induces AP-1 transcription (*Koo et al., 2020*). YAP can form a functional complex with β-catenin and TBX5 in cancer cells (*Rosenbluh et al., 2012*), YAP/TAZ can integrate into a SMAD–OCT4 complex in embryonic stem cells (*Beyer et al., 2013*), and a YAP-p73 complex plays a role in the DNA damage response (*Beyer et al., 2013*). In addition, YAP/TAZ directly interacts with the general co-activator BRD4 to increase RNA polymerase II recruitment and transcription of target genes (*Zanconato et al., 2018*), and YAP/TAZ activity is inhibited by a direct interaction with the SWI/SNF complex via the ARID1A subunit (*Chang et al., 2018*).

In previous work, we described an epigenetic switch that transforms breast cells via an inflammatory regulatory network controlled by the joint action of NF-κB, STAT3, and AP-1 transcription factors (*Iliopoulos et al., 2009*; *Iliopoulos et al., 2010*; *Ji et al., 2018*; *Ji et al., 2019*). Here, we show that YAP and TAZ are important for cellular transformation in this breast cell transformation model. YAP and TAZ directly interact with STAT3 and JUNB and act as transcriptional co-activators that stimulate expression of target genes. We provide evidence that a complex of TEAD, JUNB, and STAT3 recruits YAP/TAZ to most target sites, but that primary recruitment can be mediated by any one of these three proteins bound to their cognate motif. These different classes of YAP/TAZ target sites are associated with largely non-overlapping genes with overlapping but distinct functions. Genes containing either the AP-1 or TEAD class of YAP/TAZ sites are associated with poor survival of breast cancer patients with the triple-negative form of the disease.

## Results

### YAP and TAZ are important for STAT3 and AP-1 activity during breast epithelial cell transformation

Given the importance of YAP and TAZ in many types of human cancer, we examined their roles in breast cellular transformation by using CRISPR to knock out these genes in the context of our Src-inducible model (MCF-10A cells containing ER-Src, a fusion of the tamoxifen-inducible ligand binding domain of estrogen receptor and the v-Src oncoprotein). Treatment of these non-transformed cells with tamoxifen activates v-Src and triggers an epigenetic switch to the transformed state. Knockout of YAP or TAZ or both (*Figure 1A*) reduces growth under conditions of low attachment (*Figure 1B*) and inhibits colony formation in soft agar (*Figure 1C*). In contrast, there is no effect on cell proliferation under standard conditions of high attachment (*Figure 1D*). Similar results are observed when YAP or TAZ expression is knocked down by siRNA (*Figure 1E–G*). Thus, YAP and TAZ are important for cellular transformation in this model.

An inflammatory regulatory network mediated by the joint action of NF-κB, STAT3, and AP-1 factors at common target sites is critical for transformation in our inducible model, and this network is involved in many human cancers (*Ji et al., 2019*). Using siRNA-mediated knockdowns, we examined whether YAP and TAZ are important for the increased activity of these transcription factors during transformation. Depletion of YAP and/or TAZ, but not JUNB, strongly decreases STAT3 phosphorylation at Tyr705 (STAT3-p, a marker of STAT3 activation) during transformation (*Figure 2A*), but it does not affect overall levels of STAT3 protein (*Figure 2A*) or mRNA (*Figure 2—figure supplement 1*). Consistent with this observation, YAP and TAZ are important for induction of IL-6 transcription and secretion during transformation (*Figure 2B*). In addition, depletion of YAP or TAZ (also STAT3 and JUNB) inhibits transcriptional activation mediated by AP-1 factors (*Figure 2C*). In contrast, YAP or TAZ depletion causes a slight increase in NF-κB activation (*Figure 2D*), perhaps reflecting competition between the Hippo and NF-κB signaling pathways. The effects of YAP and TAZ on AP-1 and STAT3 activity during transformation are not due to changes in YAP or TAZ protein levels in the nucleus, which are similar in the presence of absence of tamoxifen (*Figure 3A*).

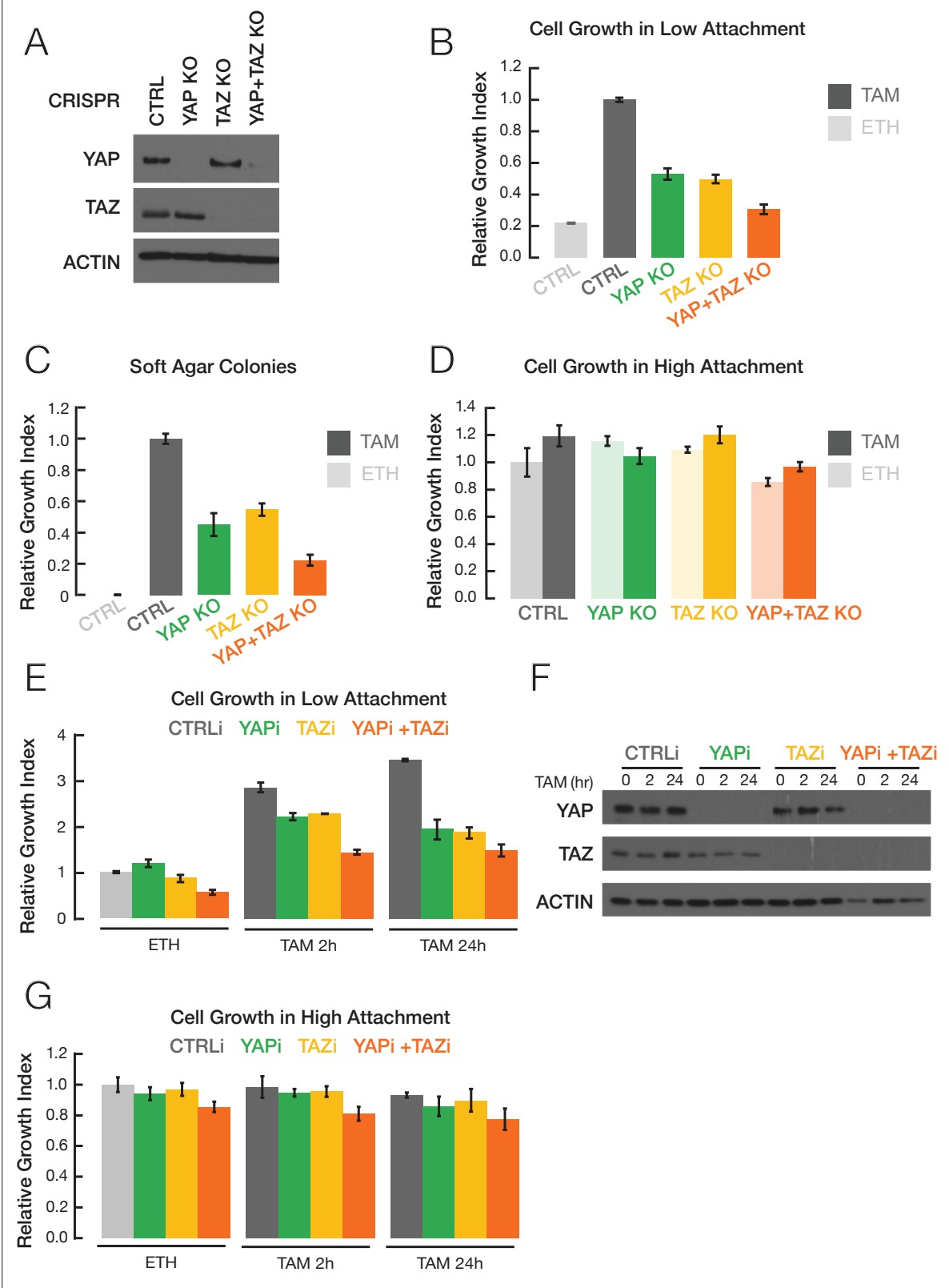

**Figure 1.** YAP and TAZ facilitate transformation. (**A**) Western blot for YAP, TAZ, and actin levels in the indicated CRISPR-mediated knockout (KO) strains and the parental cell line (CTRL). (**B**) Relative growth in low attachment conditions of the indicated cell lines in non-transformed (ETH; ethanol) and transformed conditions (TAM; tamoxifen). Measurements are relative to transformed cells with the CRISPR control, which is defined as 1.0. (**C**) Relative growth in soft agar in the indicated cell lines. (**D**) Relative growth of the indicated cell lines in standard high attachment conditions. (**E**) Relative growth in

*Figure 1 continued on next page*

*Figure 1 continued*

low attachment conditions, (**F**) YAP and TAZ protein levels, and (**G**) relative growth in high attachment conditions of cells subjected to siRNA-mediated knockdown of YAP and/or TAZ (YAPi and/or TAZi) of cells induced by TAM addition for the indicated times. Error bars indicate ± SD of 3 replicates.

## YAP and TAZ directly interact with JUNB and STAT3

The effects of YAP and TAZ knockout on STAT3 and AP-1 activity could be direct, or they could be an indirect consequence of inhibiting transformation. Among the many members of the AP-1 transcription factor family, expression of JUNB is induced during transformation (*Figure 3—figure supplement 1A*), leading us to study JUNB in more detail. We addressed the possibility of a direct effect by co-immunoprecipitation (co-IP) experiments in cytoplasmic, nucleoplasm, and chromatin fractions from non-transformed and transformed cells. YAP and TAZ proteins are observed in all three cellular fractions, and their levels are unchanged upon tamoxifen treatment (*Figure 3A*, *Figure 3—figure supplement 1B*). This indicates that the Hippo pathway is active in this cell line prior to transformation and is not altered during the transformation process. In the nucleoplasm and chromatin (but not cytoplasm) fractions, YAP and TAZ each co-IP with STAT3 and JUNB (*Figure 3B*). In general, co-IP is more efficient in transformed cells. We confirmed direct pairwise YAP-STAT3, TAZ-STAT3, YAP-JUNB, and TAZ-JUNB interactions by performing co-IP experiments using histidine-tagged recombinant proteins generated in *Escherichia coli* (*Figure 3C*).

## WW domains of YAP and TAZ are important for interacting with STAT3 and JUNB and for transformation

YAP and TAZ have a TEAD domain that interacts with TEAD1-4, and they contain WW domains (two for YAP and one for TAZ) that mediate interactions with a variety of other proteins. To map the regions of YAP and TAZ required for interacting with STAT3 and JUNB, we performed co-IP experiments in cells co-expressing YAP or TAZ derivatives lacking one or both WW domains along with FLAG-tagged STAT3 or JUNB (*Figure 3D*). The WW1 domain of YAP is critical for the interaction with STAT3 and JUNB, whereas removal of the YAP WW2 domain has no effect on these interactions. Deletion of the TAZ WW domain reduces, but does not eliminate, the interaction with either JUNB or STAT3. Removal of any WW domain has no effect on the interaction with the TEAD proteins, whereas removal of the TEAD domain abolishes the interaction with the TEAD proteins. In addition, the TEAD domains of both YAP and TAZ are critical for the interaction with JUNB, but they contribute only partially to the interaction with STAT3 (*Figure 3D*).

When overexpressed in parental MCF-10A cells (i.e., lacking the ER-Src protein), wild-type YAP or TAZ increase the level of transformation, whereas derivatives lacking YAP-WW1 or TAZ-WW do not (*Figure 3E*). Like the wild-type proteins, none of these derivatives have a significant effect on cell proliferation under conditions of high attachment (*Figure 3—figure supplement 1C*). Thus, the WW1 domain in YAP and the WW domain in TAZ, which are critical for interaction with JUNB and STAT3 but not TEAD proteins, are important for transformation. These observations also indicate that YAP and TAZ interactions with TEAD proteins are not sufficient for transformation. They are consistent with YAP and TAZ interactions with JUNB and/or STAT3 being important for transformation, although the oncogenic effects of the WW domains may involve interactions with other proteins.

## YAP and TAZ have highly similar, but not identical, genomic binding profiles

Using protein-specific antibodies, we performed ChIP-seq to identify the genomic target sites of YAP, TAZ, STAT3, JUNB, and TEAD in both non-transformed and transformed cells. In general, there is excellent agreement in genome-wide signal profiles between pairs of biological replicates for each factor (*Supplementary file 1A*), and the binding regions of the three sequence-specific transcription factors – STAT3, JUNB, and TEAD – are strongly enriched in their respective motifs (*Figure 4—figure supplement 1*). YAP, TAZ, JUNB, and TEAD have similar numbers of peaks in the non-transformed and transformed conditions, while the number of STAT3 binding sites increases fivefold upon transformation (*Supplementary file 1B*). As expected from their being paralogs, YAP and TAZ have very similar binding profiles in both the transformed and non-transformed states (*Figure 4A*; overall Pearson

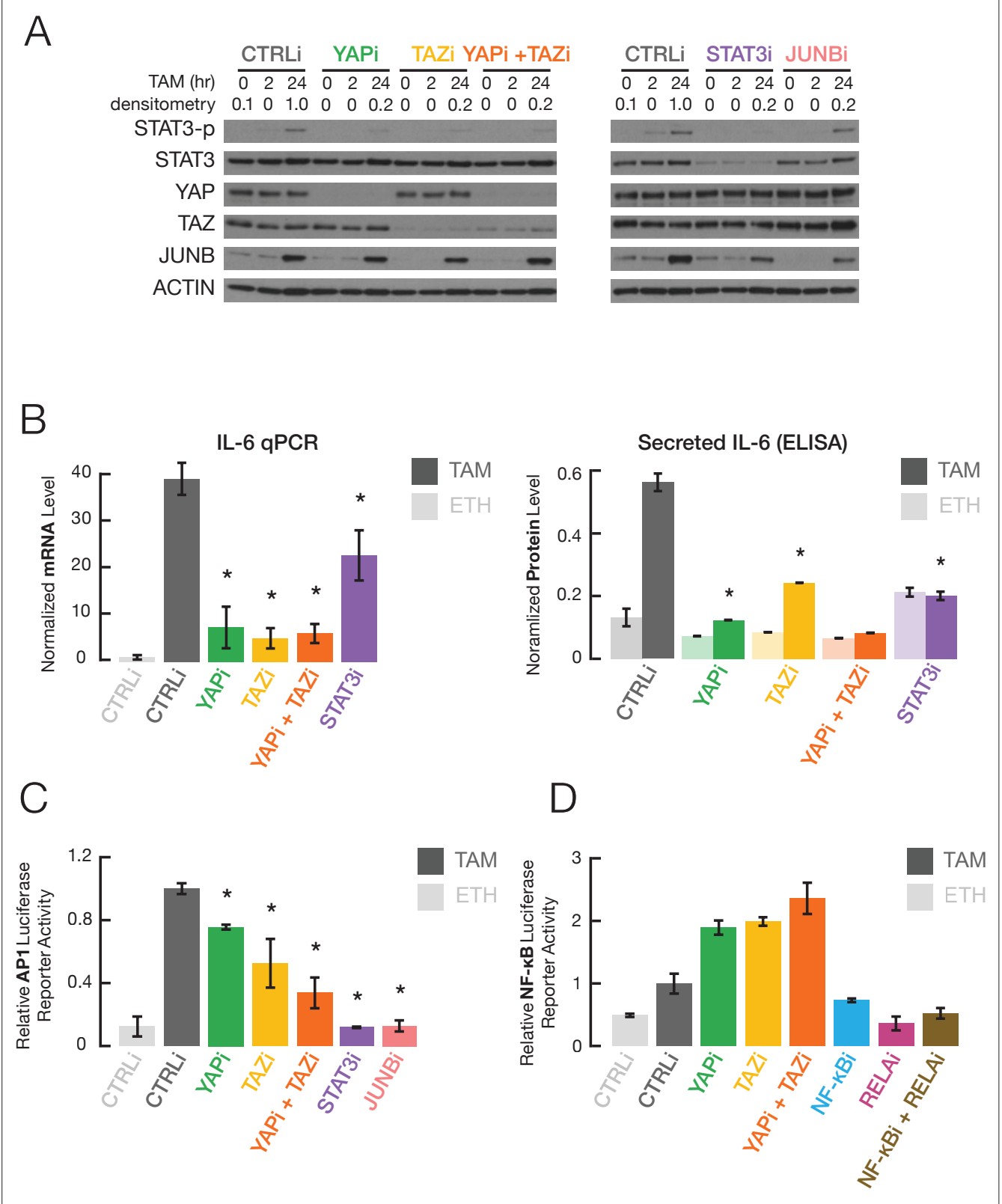

**Figure 2.** YAP and TAZ regulate STAT3 and JUNB activities during transformation. (**A**) Western blot for the indicated proteins (STAT3-p, a form phosphorylated at Tyr705) in cells treated with the indicated siRNAs for 24 hr and then with TAM for the indicated times. (**B**) Normalized IL-6 mRNA (left) and secreted IL-6 (right) levels in the indicated cells and conditions. (**C**) Relative AP-1-dependent transcriptional activity in the indicated cells and conditions. (**D**) Relative NF-$\kappa$B-dependent transcriptional activity. Error bars indicate ± SD of 3 replicates.

*Figure 2 continued on next page*

*Figure 2 continued*

The online version of this article includes the following figure supplement(s) for figure 2:

**Figure supplement 1.** Gene expression in siRNA knockdowns.

correlation coefficient ~0.7). For most subsequent analyses (but not the next section), we will consider these shared YAP/TAZ sites together.

## YAP- and TAZ-specific sites differ in their associated motifs and biological functions from shared YAP/TAZ sites

Although nearly all target sites are bound by both YAP and TAZ, ~ 575 target sites (~2 % of YAP/TAZ sites) appear specific for either YAP or TAZ (*Figure 4A*). To test whether these apparent YAP- or TAZ-specific sites are truly specific or are false positives or false negatives, we knocked out YAP or TAZ separately in ER-Src cells and then performed ChIP-seq experiments using an antibody that recognizes both proteins (*Figure 4B*). When compared to the parental cell line, a YAP-specific site should show reduced binding only in the YAP-deletion line, and a TAZ-specific site should show reduced binding only in the TAZ-deletion line. In this manner, we confirmed ~450 TAZ-specific sites and ~125 YAP-specific sites (*Figure 4C*).

Several characteristics of these YAP- and TAZ-specific sites suggest that they serve biologically distinct functions from shared sites. First, >80 % of YAP- and TAZ-specific sites are located within 150 base pairs upstream of a GENCODE-annotated transcription start site (TSS), but <25 % of shared YAP/TAZ sites are so localized (*Figure 4D*). Second, compared to shared YAP/TAZ target sites, YAP- and TAZ-specific sites are much less likely to be located within JUNB and TEAD target regions (*Figure 4E*). Third, GO analysis of the genes whose promoters are near YAP- and TAZ-specific sites identify RNA processing, and mitochondrial translation as overrepresented gene categories for TAZ-specific sites, and transcriptional regulation as categories overrepresented for YAP-specific sites (*Supplementary file 2*). Fourth, the TAZ-specific peaks are enriched in unique motifs (*Figure 4—figure supplement 1* and see below).

## YAP and TAZ co-occupy genomic target sites with STAT3, JUNB, and TEAD

We compared YAP/TAZ binding data with that of STAT3, JUNB, and TEAD proteins (*Figure 5A*). Consistent with previous results for YAP/TAZ/TEAD/AP-1 (*Zanconato et al., 2015*) and AP-1/STAT3 (*Ji et al., 2019*), YAP/TAZ, TEAD, STAT3, and JUNB have very similar binding profiles. Together with the physical interactions of YAP/TAZ with these DNA-binding proteins (*Figure 3B,C*), these observations strongly suggest that YAP/TAZ co-occupy genomic sites with STAT3, JUNB, and TEAD.

We further investigated protein co-occupancy by performing pairwise analysis of ChIP-seq peak summits in transformed cells (*Figure 5B*). Such analysis provides information on the relative locations of multiple proteins associated with genomic regions (*Wong and Struhl, 2011*; *Fleming et al., 2013*; *Petrenko et al., 2016*; *Ji et al., 2019*). For biological replicates of an individual protein, the median distances between peak summits range between 8 and 35 bp, and this serves as a control. YAP and TAZ peak summits are separated by 25 bp, indicating that these two proteins occupy the same sequences, although not necessarily at the same time. Interestingly, the median distances between pairwise combinations of YAP, TAZ, JUNB, STAT3, and TEAD range between 31 and 44 bps (*Figure 5B*), indicating that the binding locations of these proteins are very close to each other and likely coincide. Similar results are observed in non-transformed cells (*Figure 5—figure supplement 1A,B*).

We demonstrated co-occupancy of YAP and TAZ with STAT3 and JUNB by sequential ChIP experiments at selected loci. At the *IL-6* enhancer, sequential ChIP in either order yields increased fold-enrichments of YAP (*Figure 5C*) or TAZ (*Figure 5D*) with JUNB or STAT3 in non-transformed and transformed cells; similar results are observed at the *SNX24* locus (*Figure 5—figure supplement 1C*). At the *MYC* locus, increased fold-enrichment is generally seen only in one direction (JUNB or STAT3 first, except for TAZ and STAT3; *Figure 5—figure supplement 1C,D*), suggesting that most of the YAP and TAZ binding at the *MYC* locus is not mediated by JUNB or STAT3, although they can co-occupy the locus. In addition, sequential ChIP (in both orders) of JUNB and either YAP or TAZ on a transfected

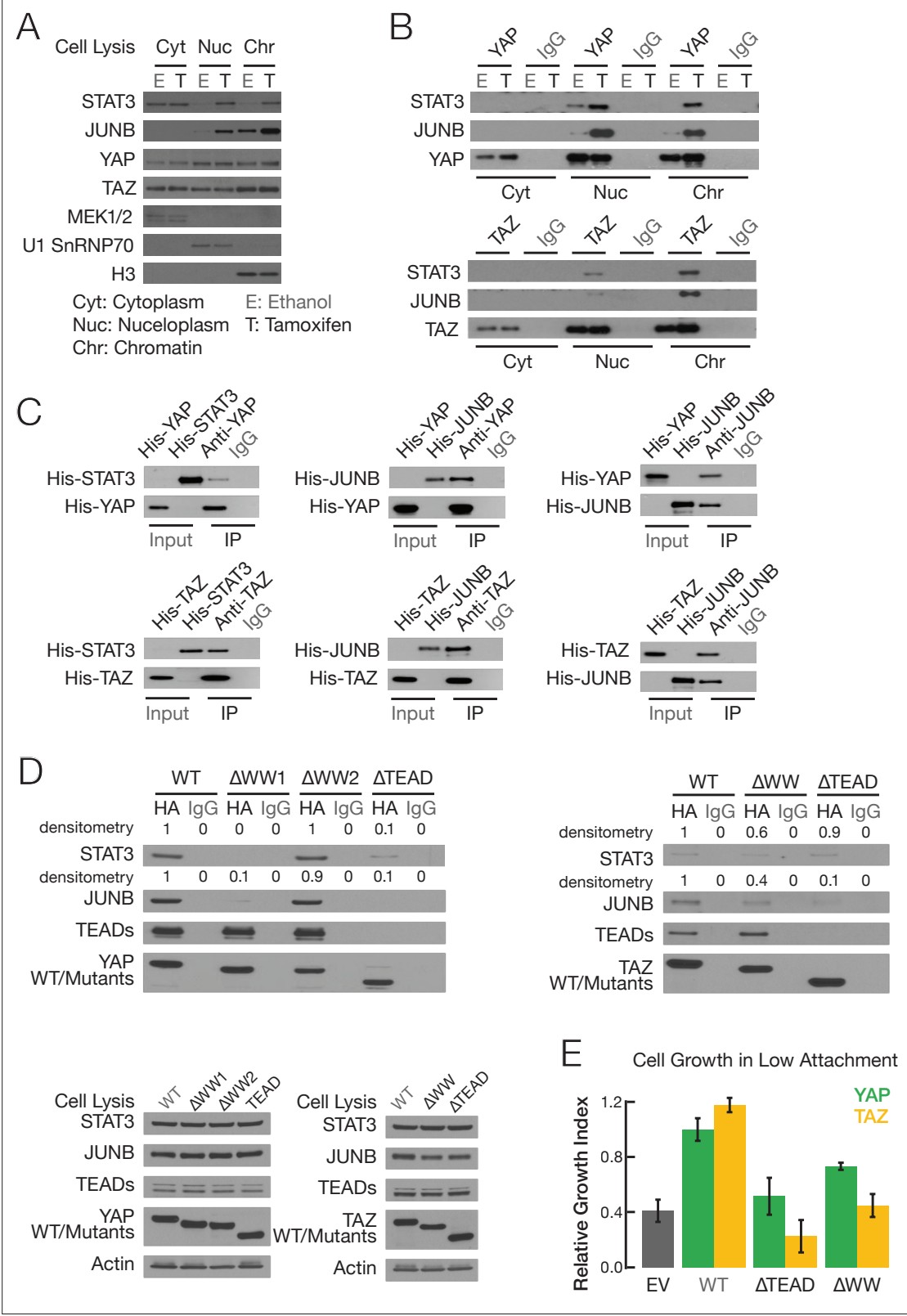

**Figure 3.** YAP and TAZ, via WW domains, directly interact with STAT3 and JUNB. (**A**) Levels of the indicated proteins in the indicated fractions in non-transformed (E; ethanol) and transformed (T; tamoxifen) conditions: MEK1/2, cytoplasm marker; U1 SnRNAP70, nucleoplasm marker; H3, chromatin marker. (**B**) Co-immunoprecipitation of endogenous proteins in cellular fractions from non-transformed and transformed cells. Western blot of the indicated proteins upon immunoprecipitation (IP) with antibodies against the indicated proteins or the IgG control. (**C**) Co-immunoprecipitation with

*Figure 3 continued on next page*

*Figure 3 continued*

*Escherichia coli*-generated His-tagged recombinant proteins. Western blot of the input and immunoprecipitated (IP) proteins with antibodies against the indicated proteins. The input sample contained 10 % of the amount used for recombinant proteins used for the co-immunoprecipitation. (**D**) Top two panels are western blots of the indicated proteins upon immunoprecipitation with the indicated HA-tagged YAP (left) or TAZ (right) derivatives or IgG control. Bottom two panels are western blots of cell extracts prior to immunoprecipitation. (**E**) Relative growth in conditions of low attachment in cells overexpressing the indicated proteins or empty vector (EV) control in parental MCF-10A cells (lacking ER-Src). Error bars indicate ± SD of 3 replicates.

The online version of this article includes the following figure supplement(s) for figure 3:

**Figure supplement 1.** Interactions of YAP and TAZ with JUNB and STAT3.

plasmid-borne locus containing six AP-1 motif sites shows increased fold-enrichment in transformed cells, while this does not occur on a control plasmid locus lacking the AP-1 sites (*Figure 5E*).

## YAP and TAZ can be recruited via AP-1, TEAD, and, to a lesser extent, STAT3 motifs

Transcriptional co-activators such as YAP and TAZ are ultimately recruited to genomic sites by sequence-specific DNA-binding proteins. Thus, motifs enriched in YAP and TAZ binding sites suggest which proteins mediate the recruitment. In ER-Src cells, JUNB binds primarily to AP-1 sequence motifs, indicative of a direct protein-DNA interaction, whereas STAT3 binding sites contain either STAT3 or AP-1 motifs, indicating that STAT3 can bind directly to DNA or indirectly via interactions with AP-1 proteins (*Ji et al., 2019*). To determine which motifs are enriched at YAP/TAZ sites, we compared the frequency of sequence motifs from the HOCOMOCO and JASPAR catalogs within YAP/TAZ target sites against comparable DNase hypersensitive sites that are not bound by YAP/TAZ.

As expected from YAP/TAZ being a co-activator of TEAD proteins, ~30 % of YAP/TAZ binding loci contain TEAD motifs (*Figure 6A*), and the TEAD motif is the most significantly enriched motif within YAP/TAZ sites (*Figure 6B*). Interestingly, and in accordance with the direct interaction of JUNB and YAP/TAZ, a comparable number of YAP/TAZ target sites contain AP-1 motifs (*Figure 6A*). However, the fold-enrichment of AP-1 motifs at YAP/TAZ sites vs. control loci is only modest (*Figure 6B*), presumably due to AP-1 factors being involved in a wide range of pathways. YAP/TAZ target sites also contain STAT3 motifs, albeit at much lower frequency (13%) and minimal enrichment over control loci. Importantly, all the TEAD, AP-1, and STAT3 motifs are centered around YAP/TAZ peak summits (*Figure 6C*), as expected for direct recruitment of YAP/TAZ by these DNA-binding transcription factors. In contrast and as a negative control, NF-κB motifs are not enriched or centered at YAP/TAZ target sites (*Figure 6A–C*).

## Different classes of YAP/TAZ target sites as defined by the recruiting motifs and protein crosslinking efficiencies

The above analysis suggests the existence of different classes of YAP/TAZ target sites, but it utilizes a stringent cutoff to define a strong motif. This leaves open the possibility that YAP/TAZ recruitment to individual loci might require multiple motifs, some of which just missed the cutoff. To address this possibility, we obtained the highest motif scores (FIMO algorithm) for AP-1, TEAD, and STAT3 motifs located ±150 bp from every YAP/TAZ peak summit. Of particular interest, we identified AP-1, TEAD, and (to a lesser extent) STAT3 classes of YAP/TAZ sites with high FIMO scores for the defining motif and low scores for the other two motifs (*Figure 6D*). In accordance with previous observations (*Zanconato et al., 2015*), some YAP/TAZ target sites contain multiple motifs (*Figure 6D*). Such composite motifs occur at a higher frequency than expected by chance (*Figure 6E*), but composite AP-1/TEAD motifs (*Zanconato et al., 2015*) only make up 9 % of the YAP/TAZ target sites, and thus are not required for YAP/TAZ recruitment. In addition, the distance between AP-1 and TEAD motifs at YAP/TAZ target sites with such composite motifs is highly variable, although some spacing relationships are slightly preferred (*Figure 6—figure supplement 1*). Thus, there are distinct classes of YAP/TAZ target sites based on the recruiting motif(s).

The observation that JUNB, TEAD, and STAT3 co-associate with distinct classes YAP/TAZ sites suggests that different forms of an AP-1/TEAD/STAT3 complex can recruit YAP/TAZ. To examine this possibility, we measured the relative crosslinking efficiencies of these DNA-binding proteins at

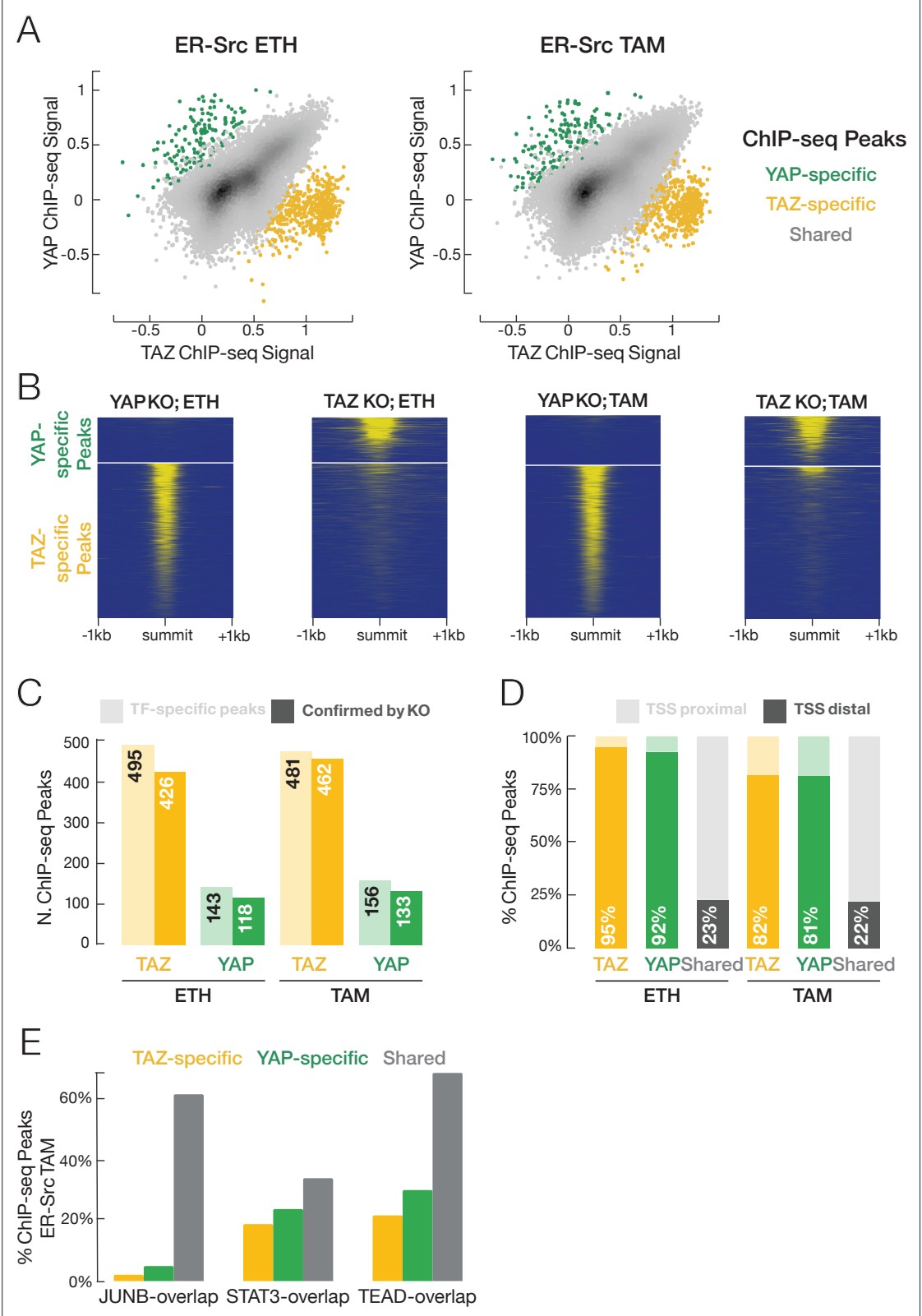

**Figure 4.** YAP and TAZ have highly similar binding profiles, but a small subset of binding sites is unique to each factor. (**A**) Correlation (r ~ 0.7) of YAP and TAZ binding signals in non-transformed (ETH) and transformed (TAM) cells. Putative YAP- and TAZ-specific sites are indicated, respectively, as green and yellow. (**B**) YAP and TAZ binding signals in YAP- or TAZ-knockout cell lines using an antibody that recognizes both proteins. (**C**) Number of putative YAP- and TAZ-specific sites (lighter colors) confirmed in cells deleted for the indicated factor (darker colors). (**D**) Percentage of YAP-specific, TAZ-specific,

*Figure 4 continued on next page*

Figure 4 continued

and shared YAP/TAZ sites that are located proximal (light colors) or distal (darker colors) to the transcription start site (TSS). (**E**) Percentages of TAZ-specific (yellow), YAP-specific (green), and YAP/TAZ-shared (gray) sites intersecting with STAT3, JUNB, and TEAD sites in transformed cells. TAZ- and YAP-specific sites are significantly less likely than shared sites to intersect JUNB and TEAD sites (Chi-square p-values YAP/JUNB = $3.2 \times 10^{-24}$, TAZ/JUNB = $2.1 \times 10^{-81}$, YAP/TEAD = 0.002, TAZ/TEAD = $9.7 \times 10^{-20}$).

The online version of this article includes the following figure supplement(s) for figure 4:

**Figure supplement 1.** Motif enrichment and overlap of ChIP-seq peaks.

different classes of YAP/TAZ sites. Strikingly, the JUNB:TEAD crosslinking ratio is strongly correlated with the quality of the AP-1 motif, whereas it is negatively correlated with the quality of the TEAD motif (*Figure 6F*). Such differential crosslinking efficiencies of transcription factors at common target sites is distinct from monolithic complexes (e.g., the Pol II preinitiation complex) where the cross-linking ratios among factors are invariant across all target sites (*Kuras et al., 2000*; *Pokholok et al., 2002*; *Rhee and Pugh, 2012*). Taken together with the direct protein–protein interactions, these results strongly suggest that, in addition to TEADs, AP-1 proteins and (to a lesser extent) STAT3 can directly recruit YAP/TAZ to genomic sites.

## CEBP motifs are enriched at YAP/TAZ target sites, and TAZ-specific sites are enriched for ETS motifs and depleted for AP-1 motifs

Unexpectedly, the motif recognized by the CEBP family of transcription factors occurs in 20 % of YAP/TAZ sites with an enrichment over control sites that is second-most behind that of TEAD motifs (*Figure 6A,B*). CEBP motifs also occur very close to YAP/TAZ peak summits (*Figure 6C*), suggesting that one or more CEBP proteins can recruit YAP/TAZ to target sites. In addition, CEBP motifs are strongly enriched within TEAD binding regions (*Figure 4—figure supplement 1*), and CEBP and TEAD motifs co-occur at YAP/TAZ sites at more than five times the rate that they co-occur in control regions (*Figure 6E*). Although CEBP proteins have not been studied in the ER-Src model, it is note-worthy that CEBPβ ranks fourth among transcription factors predicted to be important for transformation (just behind JUNB, STAT3, and FOSL1; TEAD4 ranks 15th), and siRNA-mediated depletion of CEBPβreduces the level of transformation (*Ji et al., 2018*). Lastly, a significant minority (18%) of YAP/TAZ binding sites lack AP-1, STAT3, TEAD, or CEBP motifs (*Figure 6A*), suggesting that other DNA-binding proteins can also recruit these co-activators.

Analysis of ~450 TAZ-specific sites (*Figure 4*) indicates that the aforementioned motifs are not significantly enriched (*Figure 4—figure supplement 1*, lower-right panel; there are insufficient YAP-specific sites to perform this analysis). Instead, motifs for ETS family proteins are strongly enriched and the AP-1 motif is significantly depleted at TAZ-specific sites compared to control sites. This obser-vation strongly supports the idea that TAZ-specific sites represent a biologically distinct subset from YAP/TAZ shared sites that differ with respect to the mechanism of co-activator recruitment and the genes that are affected.

## YAP and TAZ co-occupy sites with JUNB and STAT3 in a triple-negative breast cancer cell line

To provide independent support of our results, we performed ChIP-seq in a triple-negative breast cancer cell line (MDA-MB-231). As is the case in ER-Src cells, binding profiles for YAP, TAZ, STAT3, JUNB, and TEAD are extremely similar (*Figure 6—figure supplement 2A*), and there are a small number of YAP- and TAZ-specific sites (*Figure 6—figure supplement 2B*). Target sites in MDA-MB-231 cells show considerable overlap with target sites in ER-Src cells, although many sites are cell-line specific (*Figure 6—figure supplement 2C*). Pairwise analysis of ChIP-seq peak summits indicate that the binding locations of all proteins are very close and likely coincide (*Figure 6—figure supplement 2D*). The percentage of YAP/TAZ binding sites associated with AP-1, TEAD, and STAT3 motifs are comparable in both cell lines (compare *Figure 6A* with *Figure 6—figure supplement 3A,B*), and nearly half of the YAP/TAZ target sites lack these motifs. These motifs are strongly enriched very near YAP/TAZ peak summits (*Figure 6—figure supplement 3C*), and similar ratios of individual and composite motifs are observed in both cell lines (compare *Figure 6D* with *Figure 6—figure supple-ment 3D*). As in ER-Src cells, TAZ-specific sites in MDA-MB-231 cells are enriched for ELF/ETS-family

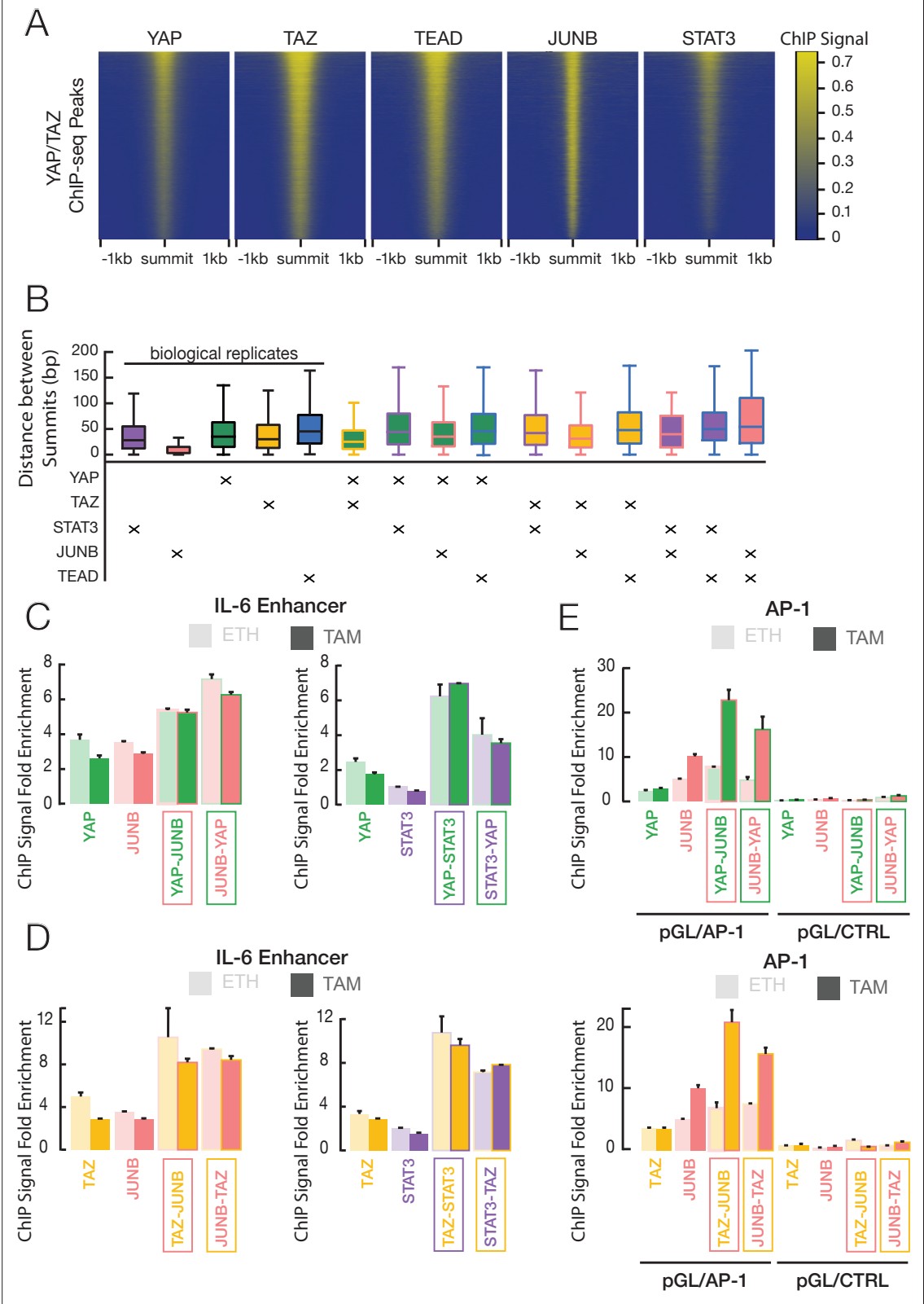

**Figure 5.** YAP and TAZ co-occupy sites with JUNB and STAT3 in transformed ER-Src cells. (**A**) Heatmaps indicate ChIP-seq signals of the indicated proteins for YAP/TAZ peaks arrayed from top to bottom in decreasing YAP ChIP signal. (**B**) Distance between peak summits for biological replicates or combinations of the indicated factors in transformed cells. (**C**) Fold-enrichments of individual and sequential ChIP at the IL-6 enhancer in untransformed (ETH) and transformed (TAM) cells with YAP and either JUNB or STAT3 performed in the indicated order. (**D**) Fold-enrichments of individual and

*Figure 5 continued on next page*

*Figure 5 continued*

sequential ChIP at the IL-6 enhancer with TAZ and either JUNB or STAT3 performed in the indicated order. (**E**) Fold-enrichments of individual and sequential ChIP of the indicated proteins on a plasmid containing either six AP-1 motifs or a control lacking these motifs (Ctrl). Error bars indicate ± SD of 3 replicates.

The online version of this article includes the following figure supplement(s) for figure 5:

**Figure supplement 1.** Co-binding of YAP/TAZ with other factors in non-transformed ER-Src cells (ethanol grown).

motifs (*Figure 6—figure supplement 3E*). Thus, the binding profiles of these factors, individually and in combination, are very similar in both cell lines.

Interestingly, there are differences between the two cell lines with respect to how YAP/TAZ is recruited to some target loci. At YAP/TAZ target sites, the enrichment of CEBP motifs in ER-Src cells is not observed in MDA-MB-231 cells (*Figure 6—figure supplement 3A,B*). Conversely, motifs recognized by the RUNX tumor suppressor are significantly enriched MDA-MB-231 cells (*Figure 6—figure supplement 3B*), but not in ER-Src cells. Thus, the putative recruitment of YAP/TAZ by CEBP proteins and RUNX appears to be cell-type dependent. This specificity could be due to differences in levels or post-translational modifications of the recruiting proteins, and it is likely to cause cell-type-specific differences in gene expression.

## Different classes of YAP/TAZ sites are associated with different categories of genes

We determined which genes are associated with the different classes of YAP/TAZ target sites in ER-Src cells (*Supplementary file 3*). For simplicity, we only considered YAP/TAZ sites with a single TEAD, AP-1, or STAT3 motif and defined an associated gene as being within 2 kb from the YAP/TAZ site. Interestingly, many genes associated with an individual motif are not associated with the other motifs (*Figure 6G*). In addition, YAP/TAZ sites differ in their GO enrichment terms based on the presumed recruiting motif (*Supplementary file 4*). YAP/TAZ sites with an AP-1 motif are enriched for genes involved in morphogenesis and cell migration, whereas sites with a TEAD motif are enriched in genes involved in actin cytoskeletal organization. No significantly enriched GO terms are observed for YAP/TAZ sites with STAT3 motifs.

## A common set of genes regulated by YAP, TAZ, STAT3, JUNB, and TEAD

To identify genes regulated by YAP, TAZ, JUNB, STAT3, or TEAD proteins, we individually depleted these factors by siRNA-mediated knockdown in tamoxifen-treated ER-Src cells. RNA-seq analysis identifies between 1000 and 4000 differentially expressed genes for each condition when compared with a control siRNA. Roughly equal number of genes show increased or decreased expression with the directionality of differential expression nearly always preserved among factors (*Figure 7A*). In accordance with previous results (*Ji et al., 2018*), there is significant overlap between the differentially expressed genes identified upon deletion of each individual factor (*Figure 7B*). Notably, binding of these transcription factors is significantly more frequent at promoters of these differentially expressed genes than at randomly generated control sets of protein-coding genes (*Figure 7C*; p-value<0.01 for all factors). However, as frequently observed (*Yang et al., 2006*), binding and transcriptional effects mediated by YAP/TAZ are discordant at many genes.

## AP-1 and TEAD classes of YAP/TAZ target genes are associated with poor prognosis in triple-negative breast cancers

The TCGA Invasive Breast Carcinoma Dataset includes 1108 breast cancer samples from 1101 patients with follow-up information, of which 116 are triple-negative breast cancers (TNBC). To determine the clinical significance of YAP/TAZ target genes in breast cancer samples, we performed Kaplan–Meier survival analysis on gene sets based on motifs at the YAP/TAZ site within 2 kb of the transcription start site of all YAP/TAZ targets. For each gene set, we computed a gene signature score (GSS) for each patient and then divided patients into low (GSS < 0) and high (GSS > 0) gene signature expression (low and high risk). Kaplan–Meier analysis was performed on the entire set of patients as well as patient subsets with luminal A, luminal B, HER2+, and TNBC cancer types.

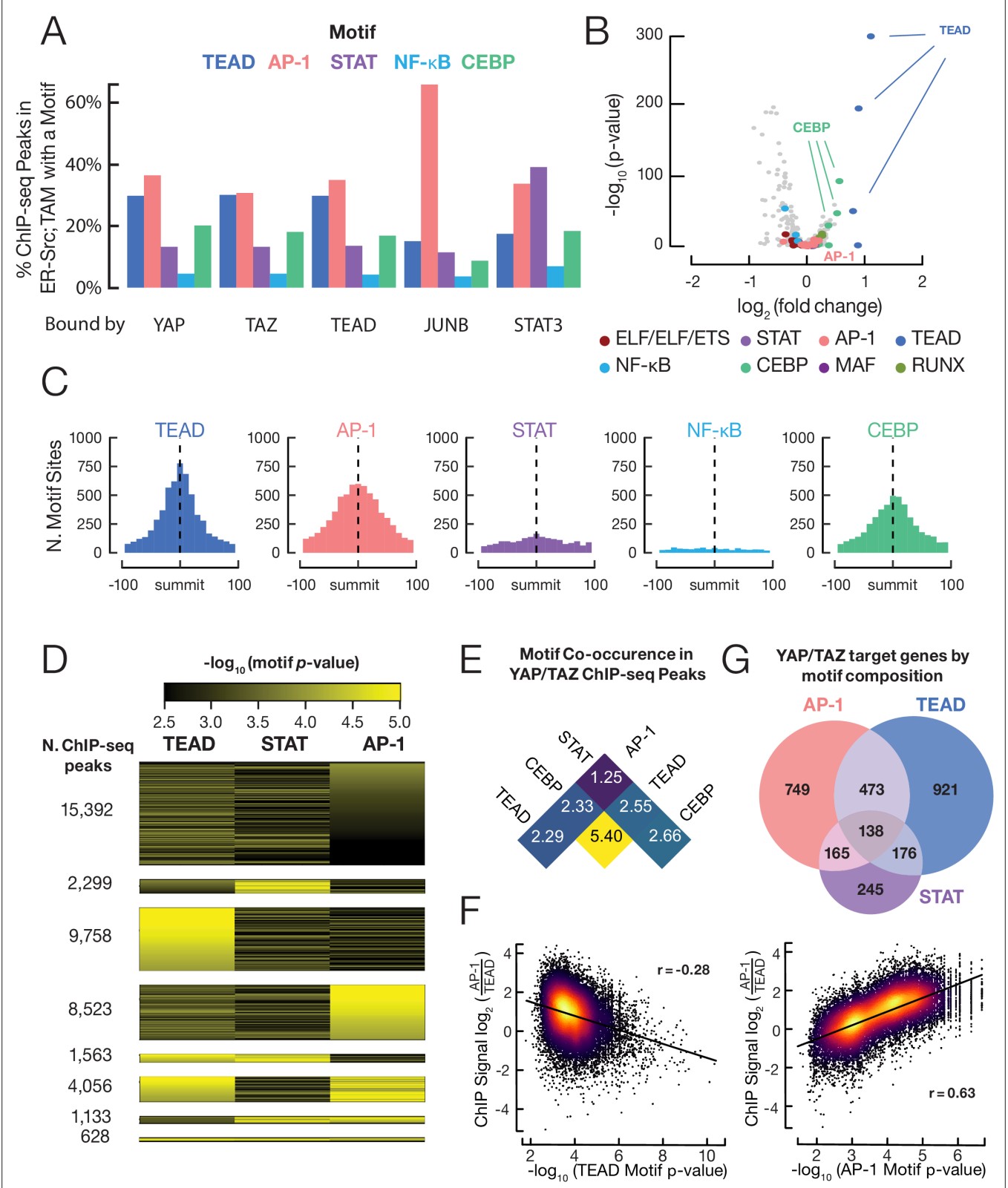

**Figure 6.** YAP/TAZ are recruited by TEAD, AP-1, CEBP proteins, and (to a lesser extent) STAT3. (**A**) Percent of binding regions for the indicated proteins that contain a given DNA sequence motif. (**B**) Enrichment and *p*-values for various motifs within YAP/TAZ target sites in transformed cells as compared with control sites having similar DNase-seq signal profiles. (**C**) Histogram of all motif locations for the indicated proteins within 500 bp of a YAP/TAZ peak summit (defined as position 0). (**D**) Numbers of YAP/TAZ peaks containing the indicated motifs and motif scores. (**E**) Fold-enrichment of the

*Figure 6 continued on next page*

Figure 6 continued

indicated pairwise combinations of motifs in YAP/TAZ target sites relative to control sites. (**F**) Scatter plots of the log-ratio of JUNB vs. TEAD ChIP-seq signals against the AP-1 motif score (top panel) and TEAD motif score (bottom panel). Pearson correlation coefficient (r) is indicated for each panel. (**G**) Venn diagram illustrating the numbers of genes with a YAP/TAZ target site classified by motif located within 2 kb of their transcription start sites.

The online version of this article includes the following figure supplement(s) for figure 6:

**Figure supplement 1.** Distribution of the distances between AP-1 and TEAD motifs at YAP/TAZ binding sites containing both motifs.

**Figure supplement 2.** YAP/TAZ binding sites and their co-binding with other factors in MDA-MD-231 cells.

**Figure supplement 3.** YAP/TAZ Motif enrichment and co-binding of YAP/TAZ with other factors in MDA-MD-231 cells.

Notably, high expression of the gene subset with YAP/TAZ sites having only AP-1 motifs is associated with significantly shorter survival in TNBC patients (*Figure 7D*; p=0.023). Similar, although slightly less significant, results are obtained for the gene subset with YAP/TAZ sites having only TEAD motifs (*Figure 7E*; p=0.06). Interestingly, high expression of the subset of genes containing YAP/TAZ sites with both AP-1 and TEAD motifs is even more significantly associated with shorter survival of TNBC patients (*Figure 7F*; p=0.002). In contrast, survival is similar for patients with high or low expression of the full 1507 YAP/TAZ target genes (*Figure 7G*; p-value=0.25). None of the gene sets examined are associated with significantly shorter survival in patients with the HER2+, luminal A, or luminal B forms of breast cancer (*Figure 7—figure supplement 1A–D*).

As YAP/TAZ target sites with both AP-1 and TEAD motifs are strongly associated with poor survival of TBNC patients, we asked whether sites with multiple AP-1 or TEAD motifs have a larger impact on survival than sites with just one motif. We defined sets of genes with proximal YAP/TAZ sites having exactly one (120 genes) or two or more (167 genes) AP-1 motifs (16 genes belong to both groups due to multiple proximal YAP/TAZ sites). We defined the same sets for TEAD motifs (292 and 104 genes, respectively, with 19 belonging to both sets). YAP/TAZ target sites with one or multiple AP-1 motif sub-signatures are associated with significant survival differences, with sites that contain multiple motifs having a slightly higher significance (*Figure 7—figure supplement 2A,B*; p=0.048 and 0.017, respectively). Both TEAD gene sets have marginal significance (*Figure 7—figure supplement 2C,D*; p=0.13 and 0.15, respectively). Thus, the very high significance of genes with YAP/TAZ sites containing both AP-1 and TEAD motifs (p=0.002) is not due to multiple motifs per se, suggesting a synergy between the AP-1 and TEAD classes of YAP/TAZ sites.

## Discussion
### YAP and TAZ are transcriptional co-activators of AP-1 proteins and STAT3

YAP and TAZ, the major effectors of Hippo signal transduction pathway (*Piccolo et al., 2014*; *Totaro et al., 2018*; *Ma et al., 2019*), play critical roles in cancer initiation, progression, metastasis, and chemo-resistance in cancer therapy (*Johnson and Halder, 2014*; *Yu et al., 2015*; *Zanconato et al., 2016*). YAP/TAZ are transcriptional co-activators that are recruited by the TEAD family of transcription factors (*Piccolo et al., 2014*; *Totaro et al., 2018*; *Ma et al., 2019*; *Moya and Halder, 2019*). As expected, YAP and TAZ are important for transformation in ER-Src cells, which is mediated by an epigenetic switch involving an inflammatory regulatory network controlled by the joint action of NF-κB, STAT3, and AP-1 transcription factors (*Iliopoulos et al., 2009*; *Iliopoulos et al., 2010*; *Ji et al., 2018*; *Ji et al., 2019*). In this cellular model, YAP and TAZ co-associate with TEAD proteins at many target sites.

Here, we provide multiple lines of evidence, demonstrating that YAP and TAZ are also transcriptional co-activators of STAT3 and AP-1 proteins. First, YAP and TAZ co-immunoprecipitate with JUNB and STAT3 in nuclear extracts. Second, all direct pairwise interactions between YAP or TAZ with JUNB or STAT3 are observed with proteins expressed in *E. coli*. The WW1 domain of YAP and the WW domain of TAZ are important for these interactions and for transformation, but they are dispensable for the interaction with TEAD proteins. Third, YAP and TAZ co-associate with JUNB and/or STAT3 at many target sites with peak summits very close together. Similar results are observed in a triple-negative breast cell line (MDA-MB-231). Fourth, many YAP/TAZ target sites coincide with AP-1 and, to a lesser extent, STAT3 motifs. Indeed, the frequency of AP-1 motifs among YAP/TAZ target sites is

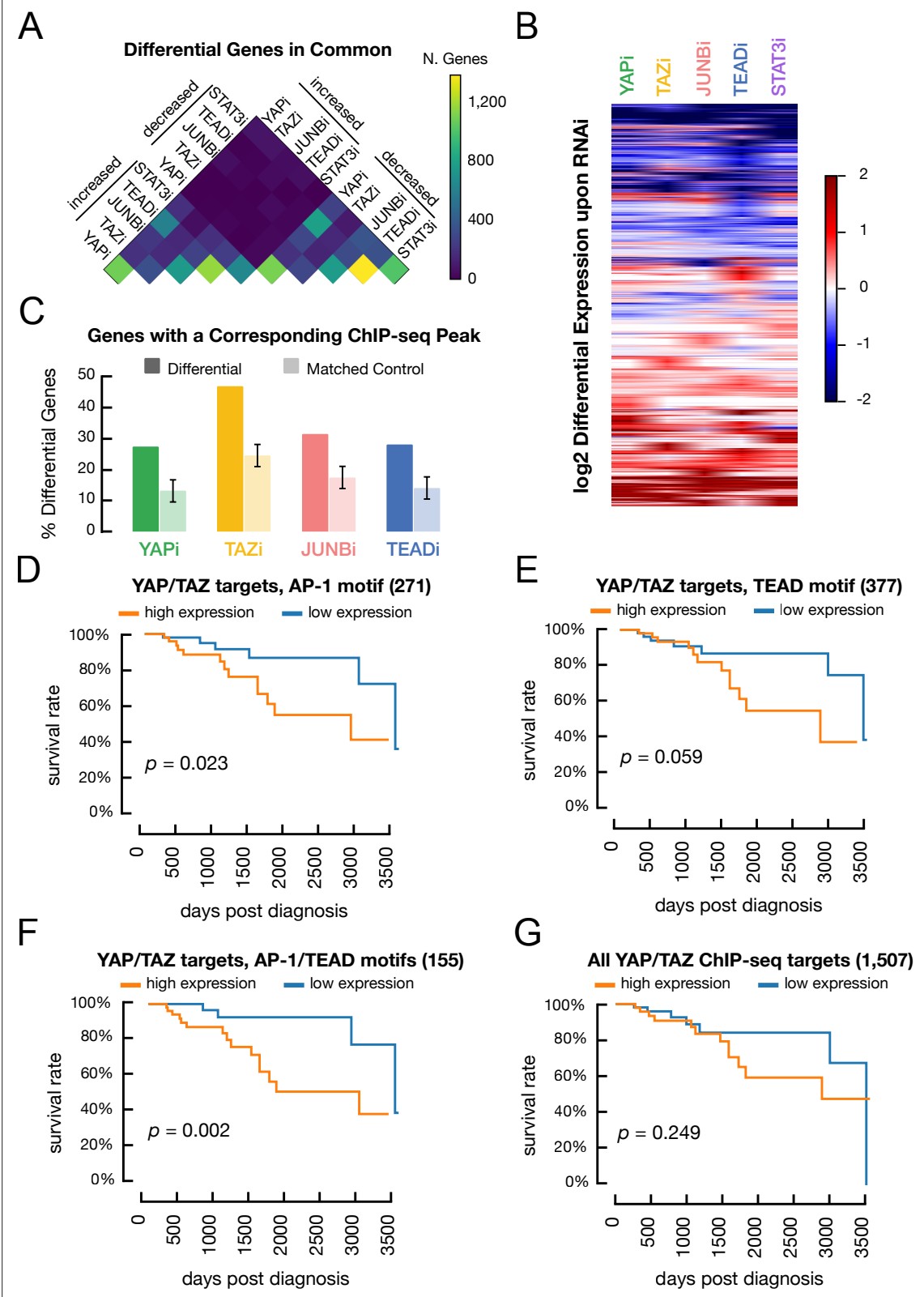

**Figure 7.** Genes regulated by YAP, TAZ, JUNB, and TEAD and association of various classes of YAP/TAZ target site with differences in overall survival in triple-negative breast cancer patients. (**A**) Overlap between genes with decreased or increased expression in transformed cells treated with RNAi against YAP, TAZ, STAT3, or JUNB when compared with a control RNAi. (**B**) Heatmap illustrates the differential gene expression (log₂) upon RNAi of each factor. Columns correspond to genes. (**C**) Percentage of differentially expressed genes following treatment with a given RNAi (dark colors) and matched

*Figure 7 continued on next page*

*Figure 7 continued*

control genes (light colors) which also have a YAP/TAZ target site for the corresponding factor within 2 kb of a transcription start site. (**D**) Kaplan–Meier survival curves for the YAP/TAZ target sites with an AP-1 motif in patients with high (orange) or low (blue) signature scores. (**E**) Kaplan–Meier survival curve for the YAP/TAZ target sites with a TEAD motif. (**F**) Kaplan–Meier survival curve for the YAP/TAZ target sites with both AP-1 and TEAD motifs. (**G**) Kaplan–Meier survival curve for all YAP/TAZ target sites.

The online version of this article includes the following figure supplement(s) for figure 7:

**Figure supplement 1.** Kaplan–Meier survival curves for the indicates genes sets from (**A**) the complete set of YAP/TAZ target sites, (**B**) the subset containing only AP-1 motifs, (**C**) the subset containing only TEAD motifs, and (**D**) the subset containing both AP-1 and TEAD motifs, for the indicated cohorts of breast cancer patients (luminal A, left; luminal B, center; HER2+, right).

**Figure supplement 2.** Kaplan–Meier survival curves for genes sets from (**A**) YAP/TAZ target sites containing exactly one AP-1 motif site, (**B**) YAP/TAZ target sites containing two or more AP-1 motif sites, (**C**) YAP/TAZ target sites containing exactly one TEAD motif site, and (**D**) YAP/TAZ target sites containing two or more TEAD motif sites.

comparable with that of TEAD motifs. Fifth, sequential ChIP experiments directly show that YAP/TAZ co-occupy target sites with JUNB and STAT3.

## Recruitment of YAP/TAZ to target sites by AP-1, STAT3, and TEAD proteins

The current model is that the YAP/TAZ co-activators are recruited by TEAD proteins that act together with AP-1 factors at genomic regions with composite sites containing AP-1 and TEAD sequence motifs (*Zanconato et al., 2015*). Consistent with this model, we confirm YAP/TAZ target sites with such composite motifs and similar DNA-binding profiles of YAP/TAZ, TEAD, and JUNB in both ER-Src and MDA-MB-231 cells. However, only a minority of YAP/TAZ target sites in both cell lines have such composite TEAD/AP-1 motifs, suggesting that additional mechanisms are involved in YAP/TAZ recruitment. In addition and consistent with previous results showing co-binding of AP-1 factors and STAT3 (*Fleming et al., 2015*; *Ji et al., 2019*), STAT3 and TEAD have similar DNA-binding profiles at YAP/TAZ sites, strongly suggesting a role for STAT3 in YAP/TAZ recruitment.

Based on our results, we propose a new model in which YAP/TAZ is recruited to target genomic regions by a complex of AP-1, STAT3, and TEAD proteins (*Figure 8*). The primary DNA-binding specificity for this complex can be individual AP-1, TEAD, or (to a lesser extent) STAT3 sequence motifs, although some genomic regions have composite motifs. This suggests that there are different classes of YAP/TAZ target sites and that the AP-1/STAT3/TEAD complex bound to DNA can exist in different forms depending on the primary DNA sequence motif. Lastly, recruitment of YAP/TAZ can involve interactions with AP-1, STAT3, TEAD, or combinations thereof.

This model is based on three lines of evidence. First, at YAP/TAZ target sites, pairwise peak analysis shows that the distances between AP-1, STAT3, TEAD, and YAP/TAZ are comparable to the distances

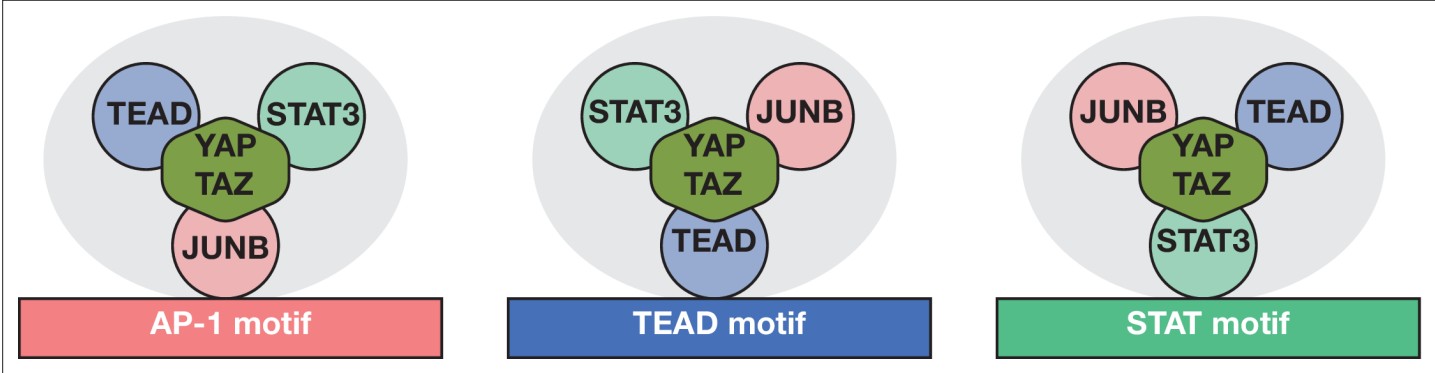

**Figure 8.** Model for YAP/TAZ recruitment by AP-1, STAT3, and TEAD proteins. Three different classes of YAP/TAZ target sites based on the recruiting motif are depicted. The overall complex (gray oval) includes all three DNA-binding proteins and perhaps additional proteins (not shown) with the cognate protein directly interacting with the indicated motif. Potential protein–DNA interactions not involving the indicated motif may exist but are not shown. Interactions between TEAD, STAT3, and AP-1 are not specified and may or may not be direct. YAP/TAZ is capable of and shown interacting with all three DNA-binding proteins, but relative contributions of the DNA-binding proteins to YAP/TAZ recruitment are not indicated or known.

between biological replicates of an individual protein. This indicates that the binding locations of all three DNA-binding proteins are very close to each other, and likely coincide, at the vast majority of genomic targets. This co-localization also suggests direct recruitment of YAP/TAZ to these target sites by one or more of the DNA-binding proteins.

Second, all three DNA-binding proteins can associate with genomic regions that have a single AP-1, STAT3, or TEAD motif, and YAP/TAZ binding peaks coincide with individual motifs. Such YAP/TAZ sites with only a single motif strongly suggest the existence of an AP-1/STAT3/TEAD complex. ChIP-seq experiments do not formally exclude the possibility that AP-1, STAT3, and TEAD occupy target regions at different times. However, this seems very unlikely at sites with only a single motif because it is hard to imagine how the non-cognate proteins would recognize the motif in the absence of the cognate protein. In addition, these considerations indicate that composite motifs are not necessary for YAP/TAZ recruitment by the AP-1/STAT3/TEAD complex. For targets with composite motifs, it is unclear whether both motifs are necessary for YAP/TAZ recruitment or whether separate AP-1/STAT3/TEAD complexes form on the individual motifs. Although not mutually exclusive, we favor the latter possibility because the distance between TEAD and AP-1 motifs is quite variable at YAP/TAZ target sites with these composite motifs.

Third, the relative crosslinking efficiencies of the DNA-binding proteins differ at the distinct classes of YAP/TAZ sites, strongly suggesting different forms of the AP-1/STAT3/TEAD complex. In contrast, monolithic complexes (e.g., the Pol II preinitiation complex and the Cyc8-Tup1 corepressor) show invariant crosslinking ratios among factors at all target sites (*Kuras et al., 2000*; *Pokholok et al., 2002*; *Wong and Struhl, 2011*; *Rhee and Pugh, 2012*; *Petrenko et al., 2019*). At an individual motif, it is virtually certain that the cognate protein directly interacts with DNA with the usual set of protein–DNA interactions necessary to generate a ChIP signal, whereas the non-cognate proteins do not. Indeed, JUNB crosslinking is relatively high at the AP-1 class of YAP/TAZ target sites, whereas TEAD crosslinking is relatively high at the TEAD class of target sites. Crosslinking of non-cognate proteins at target sites with an individual motif largely reflects protein–protein interactions, and it is possible that such proteins might not even directly interact with DNA. However, we favor the idea that the non-cognate proteins have some sequence-dependent interactions with DNA that contribute additional specificity, as this could explain why the AP-1/STAT3/TEAD complex does not interact with all individual motifs within accessible chromatin regions.

Structural details of the putative AP-1/STAT3/TEAD complex and the mechanism of YAP/TAZ recruitment remain to be elucidated. It is unknown whether AP-1, STAT3, and TEAD interact directly with each other, whether YAP/TAZ helps stabilize the complex, and whether other proteins are involved. The existence of other proteins in the complex seems possible given that some YAP/TAZ target sites lack any of these motifs. It is likely that STAT3 is not an obligate component of the complex because JUNB, TEAD, and YAP/TAZ occupancies are similar in transformed and non-transformed ER-Src cells, which have different levels of STAT3 occupancy. Finally, JUNB, STAT3, and TEAD can directly interact with YAP/TAZ, but the relative and possible synergistic contributions of the DNA-binding proteins to YAP/TAZ recruitment are unknown. Understanding these structural details will require atomic resolution of the complex on the different classes of target sites.

## Other mechanisms of YAP and TAZ recruitment

Genetic experiments indicate that YAP and TAZ have distinct and overlapping cellular functions (*Plouffe et al., 2018*; *Shreberk-Shaked et al., 2020*), but there is limited molecular understanding and no evidence for YAP- or TAZ-specific targets. Using two independent analyses, we identify 450 TAZ-specific target sites and 125 YAP-specific target sites in addition to the vastly larger number of shared YAP/TAZ sites. Interestingly, the TAZ-specific sites are enriched for ETS motifs and depleted for AP-1 motifs; TEAD motifs neither enriched nor depleted. These observations suggest an ETS protein(s) can interact specifically (although not necessarily directly) with a region of TAZ that is not found in YAP. In addition, the TAZ-specific sites are enriched as certain classes of genes, suggesting a distinct biological function.

A significant minority of YAP/TAZ target sites are not associated with AP-1, STAT3, or TEAD motifs, suggesting that other transcription factors can help recruit YAP/TAZ. CEBP motifs are enriched and frequently observed at YAP/TAZ target sites in ER-Src cells, but not in MDA-MB-231 cells. Conversely, RUNX3 motifs are enriched at YAP/TAZ target sites in MDA-MB-231 cells, but not in ER-Src cells. Such

cell-line specificity could be due to differences in the amount or activity of the recruiting protein(s) that recognize the motif and/or differences in auxiliary proteins that are important for recruitment to these motifs.

YAP/TAZ target genes associated with AP-1 or STAT3 motifs are functionally distinct, but both are associated with poor survival of patients with the triple-negative form of breast cancer.

The current view is that the biological functions of YAP/TAZ are mediated primarily through their interaction and recruitment by TEAD proteins (*Piccolo et al., 2014*; *Totaro et al., 2018*; *Ma et al., 2019*; *Moya and Halder, 2019*). However, our results indicate that a comparable number of YAP/TAZ targets involve recruitment by AP-1 proteins or (to a lesser extent) STAT3. The genes associated with these three classes of YAP/TAZ targets are largely different, and they are enriched in different functional categories. Both the AP-1 and TEAD classes of YAP/TAZ target sites are associated with poor survival of TBNC cancer patients, but little effect on survival for other forms of breast cancer. Thus, in addition to serving as a co-activator for TEAD proteins, YAP and TAZ are also co-activators for AP-1 proteins and STAT3, with both classes of target sites playing important roles in cancer.

## Materials and methods

### Cell lines and chemicals

MCF-10A-ER-Src cells (*Iliopoulos et al., 2009*; *Iliopoulos et al., 2010*; *Ji et al., 2018*; *Ji et al., 2019*) were grown in DMEM/F12 without phenol red (Thermo Fisher Scientific, 11039–047) + 5 % charcoal stripped FBS (Sigma, F6765) + 1 % pen/strep (Thermo Fisher Scientific, 15140122) + 20 ng/ml EGF (Peprotech, AF-100–15) + 0.5 µg/ml Hydrocortisone (Sigma, H-0888) + 0.1 µg/ml cholera toxin (Sigma, C-8052) + 10 µg/ml insulin (Sigma, 10516). 1–0.4 µM Tamoxifen (Sigma, H7904) + 2–4 µM AZD0530 (Selleck Chemicals, S1006) were used to induce the transformation. MDA-MB-231 cells were grown in DMEM (Thermo Fisher Scientific, 11995–073) + 10 % FBS (Sigma, TMS-013-B) + 1 % pen/strep.

### Cell transformation assays

The transformation capacity was measured by growth in low attachment conditions (GILA) (*Rotem et al., 2015*) or in soft agar. For the soft agar assay, $10^4$ cells in culture medium were mixed with 0.4 % low-melting-point agarose (VWR, 89125–532) at 37 °C and seeded on top of 1 % agarose standing layer in 12-well dishes. Colony density was measured 2–3 weeks after seeding with images captured by a digital camera (Olympus SP-350; Cam2com). For the GILA assay, 2000 cells were seeded into ultra-low attachment surface 96-well plate (Costar, 3474).  Five days after seeding, sphere cells growing in the low attachment plates were quantitated by CellTiter-Glo luminescent cell viability assay (Promega, G7571) using a SpectraMax M5 Multi-Mode Microplate Reader (Molecular Devices). Cells growing in regular culture (high attachment) dishes were stained using crystal violet, and cell density was measured using Fiji Image J's (version 1.52b) measurement function.

### CRISPR knockout and siRNA knockdown

CRISPR sgRNAs, designed with previously described algorithms (*Hsu et al., 2013*), were cloned into a CRISPR-blasticidin lentiviral plasmid, which was constructed by replacing puromycin resistant gene with blasticidin resistant gene of LentiCRISPR V2 plasmid (Addgene, #52961). The oligo sequences used to clone into CRISPR vector are as follows: YAP exon 8 – AAACTCTCATCCACACTGTTCAGGC and CACCGCCTGAACAGTGTGGATGAGA; TAZ exon 3 – AAACCCCGACGAGTCGGTGCTGGAC and CACCGTCCAGCACCGACTCGTCGGG. CRISPR lentiviral plasmids and VSV-G, GP, and REV plasmids were transfected into 293T cells to produce CRISPR lentivirus. CRISPR lentiviruses infected ER-Src cells for 1 day, and cells were selected with 10 µg/ml blasticidin (Thermo Fisher, R21001) for additional 3 days.

Oligo siRNAs were purchased from Dharmacon (siGENOME SMART pool) (*Supplementary file 5*) and were transfected into ER-Src cells using Lipofectamine RNAiMAX (Thermo Fisher Scientific, 13778050). Twenty-four  hours after transfection, cells were split and then were treated with tamoxifen and AZD0530 (4 µM) for additional 2 or 24 hours before the following assays.

## Cell fractionation, co-immunoprecipitation, and western blotting

Cell fractionation protocols were like the protocols described before (*Wu et al., 2002*) with some changes. Cells were washed with cold PBS, resuspended using buffer A (10 mM HEPES pH 7.5, 10 mM KCl and 2 mM MgCl2), lysed by grinding 50 times in Wheaton A (Wheaton, 357538), and then incubated on ice for 20 min. The lysate was placed on top of 30 % sucrose and pelleted by spinning at 15,000 RPM at 4 °C for 10 min. After removing the supernatant as the cytoplasm fraction, pellets were resuspended in buffer GB (20 mM TrisCl pH7.9, 50 % glycerol, 75 mM NaCl, 0.5 mM EDTA, and 0.35 mM DTT) and then mixed with an equal amount of buffer NLB (20 mM HEPES pH7.6, 300 mM NaCl, 7.5 mM MgCl2, 1% NP40, 1 M Urea, 0.2 mM EDTA, and 1 mM DTT) and incubated on ice for 2 min. After centrifugation at 15,000 RPM for 5 min, supernatants representing the nucleoplasm fraction were removed, and pellets resuspended using protein lysis buffer (20 mM Tris pH 7.4, 150 mM NaCl, 1 mM EDTA, 1 mM EGTA, 1 % Triton X-100, 25 mM sodium pyrophosphate, 1 mM NaF, 1 mM β-glycerophosphate, 0.1 mM sodium orthovanadate, 1 mM PMSF, 2 µg/ml leupeptin, and 10 µg/ml aprotinin). The material was sonicated 4 × 15 s using Branson Microtip Sonifier 450 at 60 % cycle duty and 4.5 output and spun at 15,000 RPM for 5 min to obtain the soluble chromatin fraction.

Co-immunoprecipitations were performed as described previously (*Iliopoulos et al., 2009*; *Iliopoulos et al., 2010*; *Ji et al., 2018*; *Ji et al., 2019*). Briefly, lysates were mixed with antibodies and 10 µl Dynabead protein G (Thermo Fisher Scientific, 10,004D) in 500 µl co-IP buffer (50 mM Tris pH 7.5, 100 mM NaCl, 1.5 mM EGTA, and 0.1 % Triton X-100), then rotated at 4 °C overnight, and washed with a co-IP buffer eight times. YAP and TAZ derivatives were cloned into pCDNA3.1 plasmid and overexpressed in 293T cells for 2 days before harvesting for co-IP experiments with anti-HA antibody or control IgG. Antibodies used for co-IP and western blot can be found in *Supplementary file 5*.

## qPCR

RNA was extracted using mRNeasy Mini Kit (Qiagen, No. 217004). One microgram RNA was converted to cDNA using SuperScript III Reverse Transcriptase (Thermo Fisher Scientific, 18080093). qPCR was running using a 7500 Fast Real-time PCR system (Applied Biosystems). qPCR primer sequences can be found in *Supplementary file 5*. All experiments were run independently three times, and values of each mRNA were normalized to that of the 36b4 internal control gene. Primer sequences are listed in *Supplementary file 5*.

## ELISA assay for IL-6 secretion

IL-6 secretion was measured using human IL-6 immunoassay kit (R&D Systems, D6050) as per manufacturer's instructions. Optical density was determined using SpectraMax M5 Multi-Mode Microplate Reader (Molecular Devices).

## Luciferase reporter assay

The pGL-AP-1 plasmid containing six consensus AP-1 binding sites was co-transfected with the pRL-CMV plasmid (Promega) into cells using TransIT 2020 transfection reagent (Mirus Bio, MIR5400). A pRL-CMV plasmid expressed Renilla protein and was used as an internal transfection control. Twenty-four hours after transfection, cells were split and treated with tamoxifen for 24 hr to induce transformation. After 3 days, firefly and Renilla luciferase activities were determined by Dual-luciferase Reporter Assay kit (Promega, E1910). To evaluate NF-κB activity, pGL4.32 (luc2P/NF-κB-RE/Hygro) (E8491, Promega) and pRL-CMV were co-transfected into cells and the same protocol followed.

## Recombinant proteins and direct interactions by co-IP

To produce recombinant proteins, YAP, TAZ, STAT3, and JUNB were cloned into pET30 plasmid. Recombinant proteins were produced in BL21 *E. coli* and dialyzed in a neutral buffer (150 mM NaCl, 46.6 mM $Na_2HPO_4$, and $NaH_2PO_4$ pH 8.0). The direct interactions among these recombinant proteins were examined in vitro using a similar co-IP procedure as mentioned above.

## ChIP-seq

Cells were dual crosslinking with a mixture of 2 mM each of ethylene glycol bis (succinimidyl succinate) and disuccinimidyl glutarate and 1 % formaldehyde. Chromatin was digested with 60 units MNase (New England Biolabs, M0247S) at 37 °C for 10 min and then sonicated using Branson Microtip

Sonifier 450 (4 × 15 s at output 4.5 and duty cycle 60%) to the sizes mostly between 150 and 500 bp. Fifty microgram chromatin, antibodies for transcription factors (listed in *Supplementary file 5*), and 15 µl Dynabead protein G (Thermo Fisher Scientific, 10,004D) was used for the chromatin immuno-precipitation (ChIP). ChIP-seq libraries were sequenced using Hiseq 2000 at the Bauer Core Facility, Harvard.

## Sequential ChIP

Sequential ChIP was performed and analyzed as described previously (*Geisberg and Struhl, 2004*; *Miotto and Struhl, 2011*). One hundred  microgram of chromatin was mixed with 20 µl Dynabead and antibodies (listed in *Supplementary file 5*) in 200 µl ChIP IP buffer (20 mM Tris–Cl pH 7.5, 140 mM NaCl, 2 mM EDTA, and 2 mM EGTA), rotated at 4 °C overnight, and sequentially washed with 1 ml each of ChIP IP buffer, ChIP wash buffer I (20 mM Tris–Cl pH 7.5, 140 mM NaCl, 2 mM EDTA, 2 mM EGTA, and 0.5 % Triton-100), ChIP wash buffer III (twice, 20 mM Tris–Cl pH 7.5, 250 mM LiCl, 2 mM EDTA, 2 mM EGTA, and 0.5 % Triton-100), and TE buffer. The bound material was eluted by 35 µl sequential ChIP elution buffer 20 mM Tris–Cl pH 7.5, 500 mM NaCl, 2 mM EDTA, 2 mM EGTA, 30 mM DTT, 0.1 % SDS, and cOmplete protease inhibitor cocktail (Roche, 11873580001) at 37 °C for 30 min. Eluted chromatin was subjected to a second ChIP by mixing with 20 µl Dynabeads and antibodies in 200 µl ChIP IP buffer and then rotated at 4 °C overnight. Chromatin from the second ChIP were washed, eluted, and de-crosslinked. Chromatin obtained from single ChIP and sequential ChIP samples were analyzed using qPCR. ChIP primers are listed in *Supplementary file 5*.

## ChIP-seq analysis

FASTQ reads were aligned to the human reference genome (GRCh38) using Bowtie2 (*Langmead and Salzberg, 2012*). For post-alignment filter steps, we used samtools-1.9 with MAPQ threshold 30 and picard-tools-2.18 to remove low-quality and duplicate reads. The SPP algorithm with `--cap-num-peak` 300,000 and IDR-2.0.4 (*Landt et al., 2012*) with `--soft-idr-threshold 0.05` were then used to determine the genomic binding sites of each transcription factor. The number of IDR peaks for each TF is listed in *Supplementary file 1*. Peaks within 2 kb of a GENCODE TSS are denoted as TSS-proximal; others are denoted as distal. DESeq2 (*Love et al., 2014*) was used to define protein-specific peaks between YAP and TAZ with the cutoff $\log_2$ fold change > 0 and multiple-testing adjusted p-value<0.05.

We performed TF occupancy analysis using their IDR peaks. We used the MACS2 pileup command to generate signal profiles which were smoothed using a 10 base pair Gaussian kernel to reduce noise. We then identified summits as the positions within a peak with the highest signal value. We adjusted IDR peaks to the summits ± 150 regions to avoid the potential bias due to varied ChIP-seq peak length distribution between transcription factors. Bedtools-2.29.2 was used to identify overlapping regions and calculate distance between peak summits.

We downloaded DNase hypersensitivity sites (DHSs) from the ENCODE Portal (*Moore et al., 2020*) for ER-Src cells treated for 24 h with tamoxifen (ENCODE accession ENCSR752EPH) and used bedtools intersect to determine that YAP/TAZ are present at 47,547 MCF10A DHSs. To assess co-occupancy of factors with YAP/TAZ for statistical significance, we compared the observed fractions of peak intersection with these YAP/TAZ-bound DHSs against a control set of 47,547 DHSs in ER-Src cells that were not bound by YAP/TAZ. We generated the control set as follows: (1) we obtained DNase-seq signal Z-scores for 2 million human DHSs from the ENCODE Encyclopedia (*Moore et al., 2020*) in tamoxifen-treated MCF10A cells (ENCODE accession ENCSR752EPH); (2) we identified DHSs from this set intersecting YAP/TAZ IDR peaks; (3) we selected 47,547 random DHSs from the ENCODE collection with the same Z-score distribution as the DHSs which intersect YAP/TAZ IDR peaks; and (4) we iteratively replaced any randomly selected DHSs which intersected YAP/TAZ peaks with alternative DHSs of the same Z-score until none of the 47,547 intersected YAP/TAZ peaks.

For de novo motif discovery, we used MEME (*Bailey et al., 2015*) to analyze sequences within 150 bp of the summits of the top 500 strongest YAP, TAZ, TEAD, JUNB, and STAT3 ChIP-seq peaks ranked by MACS2 q-value, as was done for the ENCODE Factorbook resource (*Wang et al., 2012*). We discovered five motifs for each dataset using the following MEME parameters: -dna -mod zoops -nmotifs 5 -minw 6 -maxw 30 -revcomp. We used a publicly available containerized version of the pipeline for reproducibility, downloaded from https://github.com/krews-community/factorbook-meme

(**jsonbrooks et al., 2021**). We exported the first motif discovered by MEME for the TEAD dataset, a dimeric form of the TEAD motif, for downstream analysis.

For motif occurrence analysis, TEAD, JUNB, STAT3, CEBPA, and NF-κb motifs were downloaded from the HOCOMOCO and JASPAR databases (**Kulakovskiy et al., 2018**; **Fornes et al., 2020**). We used FIMO (**Bailey et al., 2015**) to scan YAP/TAZ target sites motif matches against these and the TEAD motif identified by MEME, using a constant default FDR threshold of $10^{-4}$. We consider a peak to have a TEAD motif if it has either a TEAD monomer motif (JASPAR ID MA0090.1) or a TEAD dimer motif meeting the $10^{-4}$ threshold. For motif enrichment analysis, we downloaded the complete set of transcription factor motifs in the HOCOMOCO and JASPAR databases (**Kulakovskiy et al., 2018**) and used FIMO to scan IDR peaks for each transcription factor (**Machanick and Bailey, 2011**). We then generated control DHS sets for each set of IDR peaks as described above, scanned these sets for motif occurrences using FIMO, and computed enrichment chi-square p-values for each motif. We used this same workflow for TAZ-specific peaks.

GO analysis for YAP/TAZ-specific peaks was performed using GOrilla (**Eden et al., 2009**). Two different control gene sets were tested: (1) genes proximal to non-specific YAP/TAZ peaks and (2) genes proximal to any active DHS in MCF10A cells. The results were similar; results from the former approach are reported. For YAP/TAZ peaks separated by motif, the complete set of genes proximal to YAP/TAZ peaks was used as the control set. For non-specific YAP/TAZ peaks, the complete set of genes proximal to any active DHS in MCF10A cells was used as the control set. GO analysis for YAP/TAZ target genes grouped by motif was performed using the PANTHER gene ontology web interface (geneontology.org) with the 'biological process' category (**Mi et al., 2019**). Target genes for each motif were selected by identifying YAP/TAZ peaks containing only the corresponding motif within 2 kb of the gene's transcription start sites. PANTHER does not require a matched background gene set for comparison.

## RNA-seq analysis

RNA was prepared used mRNeasy Mini Kit (Qiagen, No. 217004). 0.4 μg total RNA was used for RNA-seq library preparation. RNA-seq libraries were generated using TruSeq Ribo Profile Mammalian Kit (Illumina, RPHMR12126). RNA-seq libraries were and sequenced at Bauer Core Facility using Hiseq 2000. For analysis, we trimmed adapter sequences, ambiguous 'N' nucleotides (the ratio of 'N' > 5%), and low-quality tags (quality score < 20). Clean reads were aligned against the GENCODE v30 reference transcriptome (**Harrow et al., 2012**) using STAR (**Dobin et al., 2013**) with the following parameters:

```
--outFilterMultimapNmax 20
--alignSJoverhangMin 8
--alignSJDBoverhangMin 1
--outFilterMismatchNmax 999
--outFilterMismatchNoverReadLmax 0.04
--alignIntronMin 20
--alignIntronMax 1000000
--alignMatesGapMax 1000000
--sjdbScore 1
```

Gene counts were normalized to TPM (transcripts per million RNA molecules) using RSEM (**Li and Dewey, 2011**) with the following parameters: '`--estimate-rspd --calc-ci`'. Differential expression analysis was performed with DESeq2 (**Love et al., 2014**) with default parameters. A separate DESeq2 run was performed for siRNA against each factor, comparing replicates treated with the siRNA against the given factor versus replicates treated with the control siRNA. DESeq2 was performed independently for tamoxifen-treated MCF10A and ethanol-treated MCF10A for each siRNA. Genes with a multiple-testing adjusted p-value < 0.01 were defined as differentially expressed.

To assess whether differential genes were significantly more likely to be directly regulated by the transcription factors of interest, we divided differential genes into two classes: TF target genes, having an IDR peak for a given factor within 2 kb of any of the gene's transcription start sites based on GENCODE v30 annotations, and non-TF target genes, not having a peak within that distance of any TSS. We compared each set of differential genes against a control set of the same number of

protein coding genes randomly selected from the GENCODE annotations. A chi-square p-value was computed for the respective fractions of target and non-target genes.

## Gene signature and survival analysis in breast cancer samples from TCGA

TCGA provisional data, including 1108 breast cancer samples, were obtained from https://www.cbio-portal.org. These samples were sub-divided into Luminal A, Luminal B, HER2+, and TNBC groups according to ER/PR/HER2 status based on immunohistochemistry (*Parker et al., 2009*; *Wallden et al., 2015*). We used the lifelines Python package to perform Kaplan–Meier survival analysis. GSS was calculated using this formula: $GSS=\sum(x_i-\mu_i)/\sigma_i$, where $x_i$ is the expression of i gene in patient samples; $\mu_i$ is mean of i gene in all patient samples; $\sigma_i$ is standard deviation. The low-expression (low-risk) group was defined as having GSS < 0, while the high-expression (high-risk) group was defined as having GSS ≥ 0 (*Adorno et al., 2009*). We refined the gene signatures by computing the coefficient of variation (CV) of the expression of all genes among these patients, removing genes with little (CV < 5%) or excessive variation (CV > 85%), the latter being likely to be statistical noise (*Mar et al., 2011*; *Jang et al., 2019*), and then filtering out pseudogenes.

In defining gene signatures, we considered an individual gene to be targeted by a transcription factor if a ChIP-seq peak for that factor fell within 2 kb of at least one of its transcription start sites from the GENCODE catalog (version 30). YAP/TAZ/JUNB and YAP/TAZ/TEAD gene signatures were defined as follows. Motif-based signatures were defined by (1) identifying all genes targeted by both YAP and TAZ in tamoxifen-treated ER-Src cells using the above criteria for ChIP-seq, then (2) filtering these to contain only genes targeted by YAP/TAZ sites containing a AP-1 motif site but not a TEAD motif site, a TEAD motif site but not a AP-1 motif site, or both a TEAD and a AP-1 motif site, using the $10^{-4}$ FIMO p-value threshold cited above. We then further divided the AP-1 and TEAD motif signatures into two sub-signatures each, the first containing genes targeted by a YAP/TAZ site having only one motif site meeting the $10^{-4}$ threshold and the second containing genes targeted by a YAP/TAZ site having two or more motif sites meeting the threshold.

## Data deposition

All sequencing data were deposited on National Cancer for Biotechnology Information Gene Expression Omnibus (GEO). GSE166943 is the accession number for all the data, with GSE166941 being the subset for the ChIP-seq data and GSE166942 for the RNA-seq data.

## Acknowledgements

We thank Zhe Ji and Xueli S Wu for help in the initial stages of bioinformatic analysis. This work was supported by grants from the National Institutes of Health to ZW (HG009446) and KS (CA 107486).

## Additional information

### Competing interests

Kevin Struhl: Senior editor, *eLife*. The other authors declare that no competing interests exist.

### Funding

| Funder | Grant reference number | Author |
|---|---|---|
| National Cancer Institute | GM 107486 | Kevin Struhl |
| National Institutes of Health | HG009446 | Zhiping Weng |

The funders had no role in study design, data collection and interpretation, or the decision to submit the work for publication.

## Author contributions

Lizhi He, Conceptualization, Data curation, Investigation, Methodology, Validation, Writing – original draft, Writing – review and editing; Henry Pratt, Mingshi Gao, Data curation, Formal analysis, Investigation, Methodology, Software, Validation, Writing – original draft, Writing – review and editing; Fengxiang Wei, Data curation, Formal analysis, Investigation; Zhiping Weng, Conceptualization, Formal analysis, Funding acquisition, Methodology, Project administration, Supervision, Writing – original draft, Writing – review and editing; Kevin Struhl, Conceptualization, Funding acquisition, Project administration, Supervision, Writing – original draft, Writing – review and editing

## Author ORCIDs

Lizhi He (ID) http://orcid.org/0000-0001-8571-3656
Mingshi Gao (ID) http://orcid.org/0000-0002-7524-892X
Zhiping Weng (ID) http://orcid.org/0000-0002-3032-7966
Kevin Struhl (ID) http://orcid.org/0000-0002-4181-7856

## Decision letter and Author response

Decision letter https://doi.org/10.7554/eLife.67312.sa1
Author response https://doi.org/10.7554/eLife.67312.sa2

---

# Additional files

## Supplementary files

• Supplementary file 1. Correlation between (A) biological replicates and (B) number of binding sites of the indicated proteins in ER-Src ethanol or tamoxifen; wild-type and YAP or TAZ knockout (KO) and MDA-MB-231 cells.

• Supplementary file 2. Enriched GO categories for YAP-specific, TAZ-specific, or shared YAP/TAZ target genes.

• Supplementary file 3. Gene signatures for the indicated categories of YAP/TAZ target sites.

• Supplementary file 4. Enriched GO terms with associated q-values for genes with YAP/TAZ target sites containing only AP-1, TEAD, or STAT motif but not combinations of motifs.

• Supplementary file 5. siRNAs, antibodies, qPCR primers, ChIP-seq primers, and list of datasets.

• Transparent reporting form

## Data availability

All sequencing data were deposited on National Cancer for Biotechnology Information Gene Expression Omnibus (GEO). GSE166943 is the accession number for all the data, with GSE166941 being the subset for the ChIP-seq data and GSE166942 for the RNA-seq data.

The following dataset was generated:

| Author(s) | Year | Dataset title | Dataset URL | Database and Identifier |
|---|---|---|---|---|
| He L, Pratt H, Wei F, Gao M, Weng Z, Struhl K | 2021 | YAP and TAZ are co-activators of AP-1 proteins and STAT3 during breast cellular transformation | https://www.ncbi.nlm.nih.gov/geo/query/acc.cgi?acc=GSE166943 | NCBI Gene Expression Omnibus, GSE166943 |

The following previously published datasets were used:

| Author(s) | Year | Dataset title | Dataset URL | Database and Identifier |
|---|---|---|---|---|
| Fleming JD, Ji Z, Struhl K | 2018 | Regulatory network controlling tumor-promoting inflammation in human cancers (ChIP-seq) | https://www.ncbi.nlm.nih.gov/geo/query/acc.cgi?acc=GSE115597 | NCBI Gene Expression Omnibus, GSE115597 |

*Continued*

| Author(s) | Year | Dataset title | Dataset URL | Database and Identifier |
|---|---|---|---|---|
| He L, Ji Z, Struhl K | 2018 | Regulatory network controlling tumor-promoting inflammation in human cancers (RNA-seq) | https://www.ncbi.nlm.nih.gov/geo/query/acc.cgi?acc=GSE115598 | NCBI Gene Expression Omnibus, GSE115598 |
| Ji Z, He L, Rotem A, Janzer A, Cheng CS, Regev A, Struhl K | 2017 | Genome-scale identification of transcription factors that mediate an inflammatory network during breast cellular transformation | https://www.ncbi.nlm.nih.gov/geo/query/acc.cgi?acc=GSE100259 | NCBI Gene Expression Omnibus, GSE100259 |
| Meuleman W | 2017 | Index and biological spectrum of human DNase I hypersensitive sites | https://www.encodeproject.org/experiments/ENCSR752EPH/ | ENCODE, ENCSR752EPH |

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
