## [Decision Letter]

**Acceptance summary:**

This work shows that effectors in the HIPPO signaling cascade function as co-activators that are recruited by transcriptional activators. Furthermore, these HIPPO signaling effectors play a role in transformation of breast epithelial cells in culture.

**Decision letter after peer review:**

Thank you for sending your article entitled "YAP and TAZ are transcriptional co-activators of AP-1 proteins and STAT3 during breast cellular transformation" for peer review at *eLife*. Your article is being evaluated by 3 peer reviewers, and the evaluation is being overseen by a Reviewing Editor and Jessica Tyler as the Senior Editor.

As you will learn from the enclosed reviews and recommendations for the authors, the enthusiasm for the article is dampened by a number of concerns in terms of novelty for the field, preliminary basis of some of the findings, and some shortcomings in experimental design and analysis.

*Reviewer #1:*

The authors have used an MCF10A-based inducible Src transformation model to study the contribution of the YAP and TAZ transcriptional coactivators to cell transformation. YAP and TAZ are transcriptional coactivators and terminal effectors of the Hippo pathway. They don't possess direct DNA binding activity, and are generally believed to be recruited to specific genomic DNA sites through "piggybacking" on TEAD family DNA binding proteins. The authors now report that YAP and TAZ can be recruited to chromatin through direct interactions with AP-1 proteins (mspecifically JunB) and STAT3. They further provide evidence that these interactions are important for cell transformation, and may be conducive to worse patient outcome in triple negative breast cancer (TNBC). They also show that while YAP and TAZ share many binding sites on the genome, each of them has also some "private" sites, associated with distinct classes of genes.

Strengths:

The findings are interesting and shed new light on transcription regulation by YAP and TAZ. In particular, they provide evidence that – at least in the setting of the studied cell transformation model – the reported interactions may be more impactful than the canonical interaction of YAP/TAZ with TEAD family members. The experiments are well performed and very clearly described.

Weaknesses:

This study has no major weaknesses. Yet, there is room for improving it. Some suggestions are listed below.

1.The data in Figure 6B raise the interesting possibility that CEBP family proteins may serve to recruit YAP/TAZ to a subset of binding sites. This is a novel and potentially valuable finding. Regrettably, the authors do not pursue it further by suitable experimental work. The authors show later that YAP/TAZ are not recruited to CEBP motifs in MDA-MB-231 cells; perhaps this putative interaction is therefore deemed by the authors irrelevant for TNBC and therefore less worthy of pursuing? It should be noted, however, that MDA-MB-231 is a mesenchymal TNBC cell line, and as such it is not representative of the majority of TNBC cases. This reviewer feels that the lack of follow-up on the CEBP lead is a missed opportunity, and such follow-up will make the paper stronger and more innovative. Do YAP/TAZ interact directly with CEBP family members? And is their binding to the indicated sites abrogated by depletion of such CEBP proteins?

2. The data presented in this study support a model where YAP and TAZ are recruited to AP-1 sites independently of TEAD. However, several earlier studies report that YAP/TAZ transcriptional interaction with AP-1 does occur in concert with TEAD (Zanconato et al., 2015; Maglic et al., 2018; Liu et al., 2016; Koo et al., 2019; Park et al., 2020). This should be discussed more thoroughly in the Discussion.

3. In view of published work by the Piccolo group showing the prominence of YAP/TAZ binding to enhancers, the authors may wish to compare the patterns of YAP/TAZ binding in concert with AP-1, STAT3 and CEBP between TSS-proximal (promoter) sites and distal enhancer sites.

4. The authors may wish to elaborate in more depth and detail on the integration of the chromatin binding site analyses and the functional implications of the transcriptional output of YAP/TAZ/JUNB vs YAP/TAZ/STAT3 vs YAP/TAZ/TEAD.

5.Please define the workflow of experiments in greater detail. For instance, in Figure 2A, how long before TAM induction of SRC was siRNA knockdown performed? Also, please define "standard conditions of high attachment". Similarly, figure legends should be more technically informative.

6. Figure 3A. According to Figure 2A, STAT3 protein levels do not increase in response to SRC induction. However, "summing up" the protein levels in "E" vs "T", it does look like STAT3 levels increase. Total lysate (input) should be presented. In either case, it would be advantageous to see subcellular localization also by immunofluorescence.

7. Was this overexpression and Co-IP experiment performed in MCF10A-ER-SRC cells induced by TAM? If yes, this should be clearly stated.

8. Line 130. "the WW domain of TAZ [is] critical for the interaction with STAT3 and JUNB" Actually, for the Co-IP experiments presented in Figure 3B and 3D, it looks like TAZ interacts only weakly (and certainly more weakly than YAP) with JUNB and STAT3. Is this the case? Is this differential YAP/TAZ behavior represented in the ChIP data? (In other words, does the YAP, compared to TAZ, chromatin binding pattern more closely overlap with JUNB or STAT3?).

9. Figure 5D and E. Is there no augmented binding of YAP/TAZ/JUNB or STAT3 on the IL6 enhancer following TAM treatment?

10. Figure 6A and line 231. It will be helpful to mention that "JUNB" in this figure is synonymous with AP-1 motif.

11. The difference and meaning of Figure 6—figure supplement 4B and 4C is not explained well enough.

12. Figure 7C is not mentioned in the text. Anyway, what does it add?

13. Figure 7E-G. To better assess the impact of YAP/TAZ on STAT3/JUNB function and their relationship to TNBC survival, Kaplan Meier plots should be generated for STAT3/JUNB alone vs STAT3/JUNB/YAP/TAZ.

14. The authors report that in addition to many target sites shared between YAP and TAZ, there also exist sites that are TAZ specific (more) or YAP specific (fewer). One might get the impression that this is an entirely novel concept. As a matter of fact, there are quite a number of earlier studies that showed differences between the transcriptional output of YAP vs TAZ in the same cell context, some including also ChIP-seq data (e.g. PMID: 26258633, 29802201, 32816858). The authors are advised to relate to that prior knowledge in their Discussion.

*Reviewer #2:*

I am not supportive of this manuscript because:

1) Above all, the message is hardly novel. There is a number of papers reporting the connections between YAP and TAZ and AP1. With experiments carried out with ChIPseq, comparably to what they did here. The field has also already offered redundant in vitro and in vivo evidence indicating that this cooperation is functional and relevant, including assays transgenic mice with phenotypes associated to YAP hyperactivation. The general conclusion from those papers is that AP1 supports YAP/TAZ function, mainly through TEAD; YAP/AP1 as a subset of YAP/TAZ biology. This is not really the case here. From this perspective, the paper is lagging behind its field.

2) A lot of emphasis is placed in 1 cell line, that is an otherwise poorly detailed Scr-inducible model. Strangely, yap/taz KO in these cells is reported to be inconsequential for cell proliferation. Why are they then using siRNA for subsequent assays?

3) The role of YAP/TAZ in inducing cell transformation, as defined by growth in soft agar or suspension is already well established. It is an assay for the field, not a result as implied here (Figure 2).

4) Transient knockdown of YAP and/or TAZ but not JUNB, strongly decreases STAT3 phosphorylation at Tyr during transformation (Figure 2A). Why is it so? this is a description, with no mechanism.

5) The biological significance of the mutual requirements shown in Figure 2 on gene expression has no biological validation. Please keep into account that YAP/TAZ knockout mice are readily available to many laboratories and commercially available; so some in vivo validation of some of these claims would not be an absurd request from the journal and an expectation of their readers.

6) In Figure 3, the association between YAP and Jun has been already described. The interaction with Stat3 is more interesting, but not developed, and with unclear biological relevance compared to what we already know, from the prior art, on the YAP-Tead and YAP/Jun association.

7) the WW Domains of YAP and TAZ are crucial for their interaction with a vast number of proteins. Concluding from their deletion that the WW domains are important for STAT2 and Jun transformation is incorrect.

8) the fact that YAP and TAZ differs in their binding to DNA may or not be relevant. There is no biology supporting that they differ in specific functions. This is again merely descriptive.

9) Conclusions (from the ER-Src cells) that TEAD is only responsible for 30% of YAP binding sites is at odd with other ChIP seq studies collectively carried out in various cell lines reporting that TEAD is the dominant factor for YAP/TAZ recruitment to DNA. There are likely technical explanations for such discrepancy, such as pull-down efficiency of various transcription factors, or relative expression of TEAD isoforms or differences in the bioinformatic pipelines. Yet the onus is on them to reconcile their work with the rest of the field.

10) in figure 6 the manuscript continues to provide descriptions with no biological explorations or even validations that the inferred gene expression relate to at least one of the many biological effects associated to YAP in vitro or in vivo.

11) Finding that a gene set for YAP/TAZ activity is able to predict outcome in breast cancer patients is just expected in light of several publications reporting the same conclusion. Showing in Fig7 and Supplement that LumA, B or HER2 show no difference in survival is at odd with a vast number of reports showing that LuminalA tumors have much more favorable outcome than other tumor types (and this is an established clinical fact), raising questions on their analyses.

*Reviewer #3:*

The Authors provide convincing biochemical data supporting the presence of intracellular complexes comprising YAP/TAZ, STAT3 and JUNB in a subset of transformed breast cancer cells.

Less clear is what is the genome-wide relationship of these complexes once associated to chromatin and what is the transcriptional relevance of these complexes, given that the genomic analyses, in their present form, do not seem to provide a clear indication of what genes and programs are controlled by these TFs, either alone or in combination.

Also, given the all the work is based on the analysis of a limited number of cell lines, the clinical relevance of the YAP/TAZ, STAT3 and JUNB complex is only inferentially supported by the predictive power of a signature of YAP/TAZ, STAT3 and JUNB regulated genes in triple negative breast cancer (but not in other breast cancer subtypes).

This may limit the generalization of their proposed model.

It is hard to make a point out of the genomic analysis presented, data is analyzed in ways that are not always canonical and which, in my opinion, prevent their interpretations (see below). Also, some conclusions do not seem to be supported by the data presented (i.e. see below the evidence showing that TAZ binding to JUNB and STAT3 is independent from its WW domain). Finally, a strong limitation of this study is the use of only two breast cell lines, one of which is a model for in vitro transformation and not a breast cancer cell line.

(1) JUNB is induced by Src, is it also required for Src dependent transformation? Based on the Authors conclusion one might expect that coregulation by YAP/TAZ/JUNB on selected genes is responsible for transformation. If so, what are the genes and programs that are directly regulated by these transcription factors (TFs)?

(2) Here are some suggestions on how to improve the analyses of genomic data (ChIP-seq):

(a) Please generate ranked heatmaps with normalized ChIP-seq signals for YAP, TEAD, TAZ, STAT3 and JUNB for the following classes of genomic sites: (i) YAP/TAZ common, YAP specific and TAZ-specific (ii) STAT3-bound peaks (iii) JUNB-bond peaks. This should be complemented by box-plot analysis of the relative enrichment signals for all the TFs in the above mentioned genomic regions.

(b) Please provide Genome browser snapshots to show quality and enrichment of TF-ChIP-seq signals on relevant genes, as canonical YAP/TAZ targets (CTGF/Cyr612), canonical STAT3 targets and canonical JUNB targets, along with representative genes co-regulated by the four transcription factors.

(3) Figure 1 reports growth assays in low attachment or low adhesion. This is meant to demonstrate that loss of YAP, TAZ or both limits growth only when cells are cultured in suspension. Text and figure legend do not report essential details as for instance at what time relative growth was measured. Therefore it is difficult to evaluate the results. Also, it is unclear which cell line was used, the Authors refer to it as "our src-inducible model", I would suggest a more informative name and reference, (for instance MFC10A-SrcER if this was the cell line used). Also, I think the formal interpretation of the results should be that YAP/TAZ are important for anchorage independent growth, rather than transformation. Some of the data presented is questionable: from figure 1E it appears that 2 hrs of tamoxifen are sufficient to lead to a three-fold increase in relative growth, (as compared to ctrl cells treated with EtOH): this seems unrealistic given that in cultures cells typically divide every 14-24 hours.

(4) Figure 1 and 2: information concerning replicates of WB experiments is missing. Also, Authors need to evaluate WB signals by densitometric analysis.

(5) The CoIP experiments shown in Figure 3A and 3B are of outstanding quality, I would ask to provide further details in the material section (number of cells used for each IP, total mg of protein used in each reaction and amount of antibody).

(6) Figure 3D. I disagree with the Authors conclusions: TAZ-DeltaWW1 still binds JUNB and STAT3 with the same efficiency of wt-TAZ as clearly visible in the panel shown here. Also, please provide information of how the coIP was performed (cell line, plasmid, protein extraction and co-IP procedure).

(7) Line 195-203: Authors used peaks summit distance to deduce whether TFs are binding to the same or different DNA sequences, for example they state that since YAP and TAZ ChIP-peaks summits are 25 bp apart, this indicates that YAZ/TAZ are binding the same DNA region. On the other hand, since the distance between STAT3 peaks (or JUNB peaks) and YAP/TAZ peaks is 30-44 bp, they deduce that these TFs are binding close but different DNA sequences. I am not aware of any metrics that allows these deductions, if these are based on prior knowledge, please cite the appropriate reference. Also, all these reasonings seem to be contradicted by later observations (line 235-237) where the Authors conclude that TEAD, AP1 and STAT3 TFs motifs are centered at YAP/TAZ peak summits.

(8) Figure 4B. color scale is missing. Also, what are the signals reported in the heatmaps? ChIP-seq signals for YAP and TAZ should be clearly indicated.

(9) Figure 4C. The description of this figure is cryptic, it is unclear what TF and KO-confirmed means. As it is, it reads as if out of the 495 TAZ specific peaks, 426 are still detected in TAZ-KO (in EtOH cells). I think the Authors mean that 426 peaks are lost in the TAZ-KO.

(10) Figure 5 A. Is this the intersection of ChIP-seq peaks? Please specify in the legend.

(11) Figure 6, suppl3: it is surprising the high number of YAP or TAZ peaks, not overlapping TEAD, this is not in line with previous observations. I ask the Authors to comment on this point. Are this really TEAD independent binding or a result of poor performance of the TEAD ChIP-seq?

(12) YAP/TAZ/STAT3/JUNB signatures are predictive of poor survival in TNBCs, what are these gene? Are they enriched in any particular pathway or process?

[Editors' note: further revisions were suggested prior to acceptance, as described below.]

Thank you for resubmitting your article entitled "YAP and TAZ are transcriptional co-activators of AP-1 proteins and STAT3 during breast cellular transformation" for peer review at *eLife*. Your article is being evaluated by 1 peer reviewer, and the evaluation is being overseen by a Reviewing Editor and Jessica Tyler as the Senior Editor.

*Reviewer #4:*

Summary

The YAP and TAZ proteins are the effectors in the HIPPO signaling cascade, serving as co-activators recruited by sequence-specific transcription factors (TFs) such as TEAD proteins. This manuscript shows that YAP and TAZ play a role in transformation of breast epithelial cells, specifically in the context of Src-inducible transformation in an MCF10A cell line. Extensive analyses of protein interactions in solution, genomic TF occupancy profiles (ChIP-seq), and differential expression upon depletion of the TFs provide new insights into the mechanisms of YAP and TAZ recruitment, adding JUNB and STAT3 as recruiters, and the potential roles of the YAP-TAZ-TF complexes recruited by different motifs.

Strengths of the manuscript:

(1) The experiments establish a role for YAP and TAZ in a specific cell transformation context.

(2) The manuscript expands and clarifies the mechanisms of recruitment of the YAP and TAZ co-activators. These co-activators are recruited not only by TEAD TFs, but also by JUNB (a subunit of AP1 family TFs) and STAT3. Previous work had shown recruitment by JUNB (AP1) in the context of a composite binding sites with TEAD TFs, and this manuscript shows recruitment by JUNB without a TEAD binding site motif. Multiple lines of evidence support this conclusion.

(3) The manuscript shows that all five (and possibly more) factors (YAP, TAZ, TEAD, JUNB, and STAT3) co-occupy many sites, both in the MCF10-inducible Src system and in a breast cancer cell line. Notably, that multi-TF complex was found at genomic intervals with binding site motif instances for single factors, indicating that the complex with recruited co-activators can be directed to distinctly different sets of targets (defined by the binding site motifs present).

(4) A series of RNA-seq determinations and analyses on cell lines with the several TFs and co-activators knocked down show that distinct sets of genes are regulated by recruitment of YAP-TAZ to the sites with each specific "recruiting motif".

(5) After defining gene sets by the apparent recruitment of the YAP-TAZ complex via AP1 or TEAD motifs, the manuscript shows that high expression of those gene sets are associated with poor prognosis in patients with triple negative breast cancer, using data from TCGA.

Weaknesses of the manuscript:

(1) While the manuscript expands our knowledge of events, recruitment, and targets in YAP-TAZ co-activation, the biological roles of the different classes of YAP-TAZ complexes and the distinct gene sets that are targets of those different classes are not explored deeply. Some enrichments for gene ontology terms are presented. I realize that this issue was brought up in the initial round of reviews, and the authors argue that further biological analyses are beyond the scope of this study. One wonders what those biological roles may be, and hopefully they will be the subject of further study.

(2) Some points in the text and panels in the figures need to be clarified. Details are in the Recommendations for the Authors.

Recommendations for the Authors

(1) Lines 96-97: The manuscript should explain the rationale for the focus on YAP and TAZ in the epithelial cell transformation system. The Introduction summarizes prior work on an epigenetic switch that drives breast cell transformation via an inflammatory network. However, no explicit statement is provided for why the Hippo pathway (YAP and TAZ) might be involved in this transformation.

(2) Lines 97-98: While the Src-inducible system has been used extensively, a brief summary would benefit readers who are not familiar with it. After reading in other sources, I learned that MCF-10A cells are not tumorigenic and presumably not transformed, and I inferred that activation of a (constitutively?) expressed ER-SRC protein will induce transformation. A brief description would be a helpful addition.

(3) Lines 118-119: The manuscript should clarify this conclusion and the data that led to it. Perhaps the key observation is that the levels of YAP and TAZ proteins in the three compartments do not change upon activation of the Src transformation protein with tamoxifen. If so, then a sentence stating that observation should be added. Then one can infer that the effects of YAP and TAZ on those AP1 and STAT3 activities just described occur without changes in the YAP or TAZ protein levels in the nucleus.

(4) Fig. 3B: Both the upper and lower sets of co-IP assay results are labeled as YAP as the antigen for immuno-precipitation. Is it possible that TAZ was the antigen for the lower set?

(5) Fig. 7B indicates some similarity in differential expression upon interference with the five factors, but it is hard to understand and it does not provide quantitative data to support the conclusion of "significant overlap". The legend say that each column corresponds to genes, but the labels on the heatmap indicate that the columns may be the differential expression in each interference condition, with expression for individual genes shown along each row. However, that should produce four discrete values along each row, but instead the color intensity changes along the row. It is not clear why the shades of red and blue change continuously if what is shown is a genes x cell condition matrix.

The number of genes in the various possible overlapping sets could be displayed in a clearer manner. For example an "upset" plot can give a readily understood, and quantitative, summary of the overlaps.

(6) Line 368 has an odd punctuation; the phrase may be "... site of all ..."

(7) Line 392: The p-values are larger than the common threshold of 0.05, and thus it would be more accurate to state something like "The results for both TEAD gene set were not significant" rather than "Both TEAD gene sets have marginal significance".

---

## [Author Response]

Reviewer #1:The authors have used an MCF10A-based inducible Src transformation model to study the contribution of the YAP and TAZ transcriptional coactivators to cell transformation. YAP and TAZ are transcriptional coactivators and terminal effectors of the Hippo pathway. They don't possess direct DNA binding activity, and are generally believed to be recruited to specific genomic DNA sites through "piggybacking" on TEAD family DNA binding proteins. The authors now report that YAP and TAZ can be recruited to chromatin through direct interactions with AP-1 proteins (mspecifically JunB) and STAT3. They further provide evidence that these interactions are important for cell transformation, and may be conducive to worse patient outcome in triple negative breast cancer (TNBC). They also show that while YAP and TAZ share many binding sites on the genome, each of them has also some "private" sites, associated with distinct classes of genes.Strengths:The findings are interesting and shed new light on transcription regulation by YAP and TAZ. In particular, they provide evidence that – at least in the setting of the studied cell transformation model – the reported interactions may be more impactful than the canonical interaction of YAP/TAZ with TEAD family members. The experiments are well performed and very clearly described.Weaknesses:This study has no major weaknesses. Yet, there is room for improving it. Some suggestions are listed below.1.The data in Figure 6B raise the interesting possibility that CEBP family proteins may serve to recruit YAP/TAZ to a subset of binding sites. This is a novel and potentially valuable finding. Regrettably, the authors do not pursue it further by suitable experimental work. The authors show later that YAP/TAZ are not recruited to CEBP motifs in MDA-MB-231 cells; perhaps this putative interaction is therefore deemed by the authors irrelevant for TNBC and therefore less worthy of pursuing? It should be noted, however, that MDA-MB-231 is a mesenchymal TNBC cell line, and as such it is not representative of the majority of TNBC cases. This reviewer feels that the lack of follow-up on the CEBP lead is a missed opportunity, and such follow-up will make the paper stronger and more innovative. Do YAP/TAZ interact directly with CEBP family members? And is their binding to the indicated sites abrogated by depletion of such CEBP proteins?

We agree that the connection of YAP/TAZ to CEBP is potentially interesting. However, additional work on CEBP is well beyond the scope of the present paper and would represent a new and major effort. First, this initial observation has nothing to do with the main conclusions of the paper, and it isn’t mentioned in the abstract. Second, to follow this up properly (i.e., not a one-off experiment with incremental value), it would require a near-equivalent amount of work to what is in the paper for AP-1 and STAT3. Third, an interesting aspect of the CEBP connection is cell-type specificity. Presumably, reviewers would want us to understand the basis of this, which is even more work. Fourth, some reviewers would devalue work on CEBP by saying that it only applies to one cell line and not another.

2. The data presented in this study support a model where YAP and TAZ are recruited to AP-1 sites independently of TEAD. However, several earlier studies report that YAP/TAZ transcriptional interaction with AP-1 does occur in concert with TEAD (Zanconato et al., 2015; Maglic et al., 2018; Liu et al., 2016; Koo et al., 2019; Park et al., 2020). This should be discussed more thoroughly in the Discussion.

As suggested, we have a much more thorough discussion of models. Indeed, we propose a new model (new Figure 8) that is clearly different from the Piccolo model.

3. In view of published work by the Piccolo group showing the prominence of YAP/TAZ binding to enhancers, the authors may wish to compare the patterns of YAP/TAZ binding in concert with AP-1, STAT3 and CEBP between TSS-proximal (promoter) sites and distal enhancer sites.

As suggested, we compare the different types of YAP/TAZ sites in terms of their location. All of them are strongly biased to distal regions, but the different classes of YAP/TAZ sites behave similarly. As shown in the original version, the YAP- and TAZ-specific sites are very strongly biased in the opposite manner, namely much more proximal than distal. This atypical behavior of YAP- and TAZ-specific sites indicates their functional differences from the shared sites.

4. The authors may wish to elaborate in more depth and detail on the integration of the chromatin binding site analyses and the functional implications of the transcriptional output of YAP/TAZ/JUNB vs YAP/TAZ/STAT3 vs YAP/TAZ/TEAD.

The relationship between protein binding and transcriptional output is shown in Figure 7C. In accord with many studies over the past 15 years, there is a clear connection between binding and transcription, but there are many cases where there is binding, but no transcriptional effect and conversely a transcriptional effect but no binding. In addition, as there are many common genes that are differentially affected by all the factors tested (also in accord with our previous results), there isn’t much more to say.

5.Please define the workflow of experiments in greater detail. For instance, in Figure 2A, how long before TAM induction of SRC was siRNA knockdown performed? Also, please define "standard conditions of high attachment". Similarly, figure legends should be more technically informative.

As stated in the methods, siRNA treatment was for 24 hr prior to adding tamoxifen for the indicated times. “High attachment” conditions are simply normal plates. Comments incorporated into methods and figure legends.

6. Figure 3A. According to Figure 2A, STAT3 protein levels do not increase in response to SRC induction. However, "summing up" the protein levels in "E" vs "T", it does look like STAT3 levels increase. Total lysate (input) should be presented. In either case, it would be advantageous to see subcellular localization also by immunofluorescence.

Total lysate was presented in Figure 2A. “Summing up” STAT3 protein levels in Figure 3A in the manner described is inappropriate because the amounts of protein loaded in the different lanes are different and are not designed to make a measurement of total STAT3, which was directly done in Figure 2A.

7. Was this overexpression and Co-IP experiment performed in MCF10A-ER-SRC cells induced by TAM? If yes, this should be clearly stated.

Co-IP experiments in Figure 3A involve endogenous proteins; no overexpression is involved. We clarified this in the figure legend.

8. Line 130. "the WW domain of TAZ [is] critical for the interaction with STAT3 and JUNB" Actually, for the Co-IP experiments presented in Figure 3B and 3D, it looks like TAZ interacts only weakly (and certainly more weakly than YAP) with JUNB and STAT3. Is this the case? Is this differential YAP/TAZ behavior represented in the ChIP data? (In other words, does the YAP, compared to TAZ, chromatin binding pattern more closely overlap with JUNB or STAT3?).

The co-IP experiments involving YAP vs. TAZ are not suitable for quantitation of interaction strength with JUNB and STAT3 for many reasons, and differences in band intensities are subtle in any event. Similarly, ChIP experiments can’t measure absolute differences between YAP and TAZ binding, but the results are remarkably similar (correlation ~ 0.7), both qualitatively and quantitatively. Indeed, for most analyses, we combine shared YAP and TAZ sites (98% of all sites) for simplicity.

9. Figure 5D and E. Is there no augmented binding of YAP/TAZ/JUNB or STAT3 on the IL6 enhancer following TAM treatment?

In Figure 5D, E (now C, D), the results for YAP, TAZ, and JUNB are expected from the genome-wide ChIP analysis showing a similar number of sites with similar binding signals. We don’t have a good explanation for why STAT3 binding doesn’t increase upon tamoxifen.

10. Figure 6A and line 231. It will be helpful to mention that "JUNB" in this figure is synonymous with AP-1 motif.

We clarify that the JUNB and AP-1 motif is the same.

11. The difference and meaning of Figure 6—figure supplement 4B and 4C is not explained well enough.

We agree that Figure 6-supplement 4B, C was confusing and simply eliminated it, as it had minimal value.

12. Figure 7C is not mentioned in the text. Anyway, what does it add?

Figure 7C was mentioned in the original text, but it was rather opaque. It has been replaced by new Figure 7B, a histogram of transcriptional profiles upon siRNA knockdowns, to make the point that the various knockdowns give very similar transcriptional profiles. This result is similar to that in our previous paper (Ji et al., 2018) that involved multiple DNA-binding transcription factors; YAP/TAZ behave similarly.

13. Figure 7E-G. To better assess the impact of YAP/TAZ on STAT3/JUNB function and their relationship to TNBC survival, Kaplan Meier plots should be generated for STAT3/JUNB alone vs STAT3/JUNB/YAP/TAZ.

As suggested, we have performed independent Kaplan-Meyer analyses for the various classes of YAP/TAZ sites.

14. The authors report that in addition to many target sites shared between YAP and TAZ, there also exist sites that are TAZ specific (more) or YAP specific (fewer). One might get the impression that this is an entirely novel concept. As a matter of fact, there are quite a number of earlier studies that showed differences between the transcriptional output of YAP vs TAZ in the same cell context, some including also ChIP-seq data (e.g. PMID: 26258633, 29802201, 32816858). The authors are advised to relate to that prior knowledge in their Discussion.

Regarding YAP- and TAZ-specific sites, this is a novel conclusion. See point 3 in the novelty/advance section for a more detailed explanation.

Reviewer #2:

Review 2 is replete with inconsistencies, bias, and negative innuendo (particularly comments 1-4, 9); it should be discarded. Though blunt, this rebuttal will focus solely on the statements in the actual review. It is not a personal attack on Reviewer 2, and I don’t believe that Review 2 is a personal attack on me.

I am not supportive of this manuscript because:1) Above all, the message is hardly novel. There is a number of papers reporting the connections between YAP and TAZ and AP1. With experiments carried out with ChIPseq, comparably to what they did here. The field has also already offered redundant in vitro and in vivo evidence indicating that this cooperation is functional and relevant, including assays transgenic mice with phenotypes associated to YAP hyperactivation. The general conclusion from those papers is that AP1 supports YAP/TAZ function, mainly through TEAD; YAP/AP1 as a subset of YAP/TAZ biology. This is not really the case here. From this perspective, the paper is lagging behind its field.

This “above all, the message is hardly novel” criticism is incorrect and illogical. We agree that “there are a number of papers reporting connections between YAP/TAZ and AP-1”, and indeed we cited the Piccolo work (and now cite additional papers mentioned in Review 1). However, the YAP/TAZ/AP-1 connection per se is NOT the message of our paper. Instead, the novelty of our work is mechanistic, namely a new molecular model (direct co-activation by AP-1 and STAT3) and different modes of YAP/TAZ recruitment that doesn’t involve TEAD binding to its motifs. Indeed, Review 2 agrees with us by saying that the current view is that “AP-1 supports YAP/TAZ function, mainly through TEAD”, and that “this is not really the case” in our paper. It is illogical to claim that our paper is “hardly novel” yet doesn’t agree with the conventional view. And how can a paper that makes novel conclusions that differ from the conventional view be viewed as “lagging behind the field”? This misrepresentation of “our message” permeates the entire review.

2) A lot of emphasis is placed in 1 cell line, that is an otherwise poorly detailed Scr-inducible model. Strangely, yap/taz KO in these cells is reported to be inconsequential for cell proliferation. Why are they then using siRNA for subsequent assays?

The statement that we use a “poorly detailed Src-inducible model” is unwarranted as we have published 26 papers using this model, resulting in >5000 citations. Furthermore, this gratuitous comment has nothing to do with the actual results or conclusions of the paper; it is just negative innuendo.

On a different point, it is not “strange” that we do siRNA experiments instead of knockouts. Knockouts involve long-term culture, and cells often adapt to the new genetic condition. siRNA (and other depletion) experiments are better for looking at more immediate effects of the genetic perturbation, even though depletions are not complete. The two approaches are not equivalent, and both have value. For example, in our work on yeast Cyc8-Tup1 co-repressor (Wong and Struhl, 2011 Genes Dev), depletion experiments were critical for elucidating the molecular mechanism that differed from the previous models that were based on knockouts.

3) The role of YAP/TAZ in inducing cell transformation, as defined by growth in soft agar or suspension is already well established. It is an assay for the field, not a result as implied here (Figure 2)

It is well known that YAP/TAZ is important for cell transformation and cancer; that is why we were interested in its role in our model. Figure 1 is simply the obvious control showing that YAP/TAZ is important in our model. To imply that we claim this as novel and an important result of the paper is negative innuendo.

4) Transient knockdown of YAP and/or TAZ but not JUNB, strongly decreases STAT3 phosphorylation at Tyr during transformation (Figure 2A). Why is it so? this is a description, with no mechanism.

The fact that transient knockdown of YAP and/or TAZ on STAT3 phosphorylation (Figure 2) is simply to show a connection between YAP/TAZ and STAT3. This was hardly a major point of the paper, and we never emphasized it. To imply that we claim this as an important result that is critical for any major conclusion of the paper is negative innuendo.

5) The biological significance of the mutual requirements shown in Figure 2 on gene expression has no biological validation. Please keep into account that YAP/TAZ knockout mice are readily available to many laboratories and commercially available; so some in vivo validation of some of these claims would not be an absurd request from the journal and an expectation of their readers.

As explained above, this paper is concerned with mechanism, not biology, so the issue raised is irrelevant to the central conclusions. It is also unclear what experiment is being suggested (other than a vague statement about using knockout mice). Moreover, it is highly likely that results with knockout mice will vary considerably, depending on which cell types are analyzed and what experimental conditions are chosen. By this logic, the mouse knockout experiments will either validate or invalidate the results in our model depending on which cell types and conditions are used. Also, as discussed before, all the biological experiments performed by others are equally consistent with our model and the Piccolo model.

6) In Figure 3, the association between YAP and Jun has been already described. The interaction with Stat3 is more interesting, but not developed, and with unclear biological relevance compared to what we already know, from the prior art, on the YAP-Tead and YAP/Jun association

Perhaps I’m missing something, but I’m unaware of a direct YAP/TAZ interaction with Jun (or other AP-1 family member) or STAT3. This would require the use of purified proteins in addition to co-IP, both of which are presented in our paper. Furthermore, multiple additional lines of evidence indicate that the YAP/TAZ interactions with AP-1 and STAT3 can be independent of TEAD binding to its cognate motif. How else does one explain 1) the YAP/TAZ target sites that lack TEAD motifs, 2) the importance of YAP/TAZ on transcriptional activation on an AP-1 reporter construct (lacks TEAD motifs), and 3) YAP/TAZ binding and co-binding with JunB (sequential ChIP) to an AP-1 reporter? These results are not explained by the Piccolo model.

7) The WW Domains of YAP and TAZ are crucial for their interaction with a vast number of proteins. Concluding from their deletion that the WW domains are important for STAT2 and Jun transformation is incorrect.

We are aware that the WW domains interact with multiple proteins and hence that the WW deletion results are only consistent with, not proof, that the interactions with JunB and STAT3 are important for transformation. We have further softened the wording in the manuscript, and never claimed proof. It should be noted that the WW domain deletions still interact with TEAD at normal levels, yet these derivatives are defective in transformation. This indicates that the TEAD interaction is not sufficient for transformation, thereby indicating that other interactions of the WW domain are important. This is a minor aspect of the paper.

8) The fact that YAP and TAZ differs in their binding to DNA may or not be relevant. There is no biology supporting that they differ in specific functions. This is again merely descriptive.

It is unclear what is meant by “biology”. However, aside from demonstrating YAP- and TAZ-specific direct targets (novel; see point 3 in previous section), we provide strong evidence for distinct functions of these specific sites. First, YAP- and TAZ-specific sites are associated with specific, yet different, classes of genes. Second, TAZ-specific sites are enriched for unique motifs suggesting that they are recruited by different transcription factors, which of course have different functions. Third, YAP- and TAZ-specific sites are highly biased to proximal locations, the opposite of YAP/TAZ sites that are highly biased to distal sites. These specific functions occur far beyond random expectation and hence are evolutionarily selected, so it is highly unlikely that they are “irrelevant”.

9) Conclusions (from the ER-Src cells) that TEAD is only responsible for 30% of YAP binding sites is at odd with other ChIP seq studies collectively carried out in various cell lines reporting that TEAD is the dominant factor for YAP/TAZ recruitment to DNA. There are likely technical explanations for such discrepancy, such as pull-down efficiency of various transcription factors, or relative expression of TEAD isoforms or differences in the bioinformatic pipelines. Yet the onus is on them to reconcile their work with the rest of the field.

The comment that “the onus is on us to reconcile our work with the rest of the field” is anti-scientific. This medieval viewpoint would block new findings and allow published work to go unchallenged. On the contrary, the onus is on Reviewers to review the paper at hand. If they disagree with the conclusions, they must make reasoned arguments based on what is presented. Review 2 doesn’t do this at all.

Furthermore, the specific comments about pull-down efficiency, TEAD isoforms, and bioinformatic pipelines do not make sense, because the key observation is based on motifs, not JUNB and TEAD binding. The empirical result that only 30% of YAP/TAZ target sites have TEAD motifs is based on ChIP-seq data with 3 independent antibodies (YAP, TAZ, dual YAP/TAZ) that give near-identical results. Is Review 2 suggesting that 30% of our data is correct and 70% is incorrect? Or is Review 2 suggesting that we don’t know how to perform and analyze ChIP-seq experiments and do motif analysis?

Lastly, the comment is incorrect because the paper explicitly says that we find examples that fit the Piccolo model of composite sites. However, the results clearly show that most YAP/TAZ sites do not have TEAD motifs and hence cannot be explained by the Piccolo model. More generally, comment 9 is strongly suggestive of bias, which (along with other comments) is why I request that this review be discarded.

10) in figure 6 the manuscript continues to provide descriptions with no biological explorations or even validations that the inferred gene expression relate to at least one of the many biological effects associated to YAP in vitro or in vivo.

I don’t understand what “no biological explorations or even validations that the inferred gene expression relate to at least one of the many biological effects” is supposed to mean. Figure 6 is concerned with motif analysis of YAP/TAZ target sites, a mechanistic issue totally unrelated to this comment.

11) Finding that a gene set for YAP/TAZ activity is able to predict outcome in breast cancer patients is just expected in light of several publications reporting the same conclusion. Showing in Fig7 and Supplement that LumA, B or HER2 show no difference in survival is at odd with a vast number of reports showing that LuminalA tumors have much more favorable outcome than other tumor types (and this is an established clinical fact), raising questions on their analyses.

Review 2 misses the point about the survival analysis (Figure 7D-G). This novelty of this analysis is that it specifically examines different classes of YAP/TAZ target genes, something that could not be done previously because such target genes are identified in our paper. This analysis provides biological importance to the molecular mechanism (direct co-activation via AP-1 and STAT3) that is the central conclusion of the paper. Thus, our results are new and hence not “just expected” or “at odds with a vast number of reports” (how could they be both?). If Review 2 wishes to question our standard analysis, it is necessary to address the analysis itself. Lastly, the “established clinical fact” that luminal A tumors have more favorable outcome than other tumor types is unrelated to our analysis and hence irrelevant. We are not comparing tumor types for patient outcomes, but rather whether a set of molecularly defined genes is linked to patient outcomes.

Reviewer #3:The Authors provide convincing biochemical data supporting the presence of intracellular complexes comprising YAP/TAZ, STAT3 and JUNB in a subset of transformed breast cancer cells.Less clear is what is the genome-wide relationship of these complexes once associated to chromatin and what is the transcriptional relevance of these complexes, given that the genomic analyses, in their present form, do not seem to provide a clear indication of what genes and programs are controlled by these TFs, either alone or in combination.Also, given the all the work is based on the analysis of a limited number of cell lines, the clinical relevance of the YAP/TAZ, STAT3 and JUNB complex is only inferentially supported by the predictive power of a signature of YAP/TAZ, STAT3 and JUNB regulated genes in triple negative breast cancer (but not in other breast cancer subtypes).This may limit the generalization of their proposed model.General Comments: It is hard to make a point out of the genomic analysis presented, data is analyzed in ways that are not always canonical and which, in my opinion, prevent their interpretations (see below). Also, some conclusions do not seem to be supported by the data presented (i.e. see below the evidence showing that TAZ binding to JUNB and STAT3 is independent from its WW domain). Finally, a strong limitation of this study is the use of only two breast cell lines, one of which is a model for in vitro transformation and not a breast cancer cell line.

See comment 4 in the novelty/advance section about the predominant use of a single cell line.

(1) JUNB is induced by Src, is it also required for Src dependent transformation? Based on the Authors conclusion one might expect that coregulation by YAP/TAZ/JUNB on selected genes is responsible for transformation. If so, what are the genes and programs that are directly regulated by these transcription factors (TFs)?

In our model, JUNB is one of many examples where the gene is induced by Src (and Ras and other stimuli) and important for transformation. This has been extensively studied in many of our previous papers, and indeed is the basis of the epigenetic switch that relies on an inflammatory positive feedback loop. The present paper doesn’t add much on the inflammatory loop *per se*, but various classes of YAP/TAZ target genes are discussed in the paper.

(2) Here are some suggestions on how to improve the analyses of genomic data (ChIP-seq):(a) Please generate ranked heatmaps with normalized ChIP-seq signals for YAP, TEAD, TAZ, STAT3 and JUNB for the following classes of genomic sites: (i) YAP/TAZ common, YAP specific and TAZ-specific (ii) STAT3-bound peaks (iii) JUNB-bond peaks. This should be complemented by box-plot analysis of the relative enrichment signals for all the TFs in the above mentioned genomic regions.(b) Please provide Genome browser snapshots to show quality and enrichment of TF-ChIP-seq signals on relevant genes, as canonical YAP/TAZ targets (CTGF/Cyr612), canonical STAT3 targets and canonical JUNB targets, along with representative genes co-regulated by the four transcription factors.

Best reviewer comment I’ve received in years. We did the suggestion as well as follow-up analyses, and obviously, the manuscript has changed for the better, including a different (and novel) molecular view of what is going on. We don’t think a Gene Browser snapshot for a single locus adds much given the new Figure 5A, which is far more relevant and informative.

(3) Figure 1 reports growth assays in low attachment or low adhesion. This is meant to demonstrate that loss of YAP, TAZ or both limits growth only when cells are cultured in suspension. Text and figure legend do not report essential details as for instance at what time relative growth was measured. Therefore it is difficult to evaluate the results. Also, it is unclear which cell line was used, the Authors refer to it as "our src-inducible model", I would suggest a more informative name and reference, (for instance MFC10A-SrcER if this was the cell line used). Also, I think the formal interpretation of the results should be that YAP/TAZ are important for anchorage independent growth, rather than transformation. Some of the data presented is questionable: from figure 1E it appears that 2 hrs of tamoxifen are sufficient to lead to a three-fold increase in relative growth, (as compared to ctrl cells treated with EtOH): this seems unrealistic given that in cultures cells typically divide every 14-24 hours.

We have described and used the low-attachment assay is multiple papers, but as suggested, we add more methodological details. Growth in soft agar has been the standard transformation assays for 6 decades, and our low attachment assay is just a different version of the same assay (both are presented). In addition, we clarify the 2-hr experiment. After the 2-hr tamoxifen treatment, the cells are seeded on low attachment plates (thereby removing the tamoxifen) and cells grow for 4 days before being assayed. The Reviewer may not be aware of our initial paper on this system (Iliopoulos et al., 2009 Cell) showing that short-term tamoxifen treatment is sufficient for transformation (it can even be much shorter than 2-hr, although that increases the time necessary for transformation).

(4) Figure 1 and 2: information concerning replicates of WB experiments is missing. Also, Authors need to evaluate WB signals by densitometric analysis.

As suggested, we have quantitated WB signals.

(5) The CoIP experiments shown in Figure 3A and 3B are of outstanding quality, I would ask to provide further details in the material section (number of cells used for each IP, total mg of protein used in each reaction and amount of antibody).

We appreciate the comment about outstanding quality. Requested details (5 million cells/IP; amount of antibody listed in Table) provided.

(6) Figure 3D. I disagree with the Authors conclusions: TAZ-DeltaWW1 still binds JUNB and STAT3 with the same efficiency of wt-TAZ as clearly visible in the panel shown here. Also, please provide information of how the coIP was performed (cell line, plasmid, protein extraction and co-IP procedure)

The reviewer is partially correct. Unlike what we implied in the original paper, the TAZ-WW derivative does interact with JUNB and STAT3, but quantitation (now provided) indicates less well than wild-type TAZ. We have modified the text. All YAP and TAZ derivatives were cloned into pCDNA3.1 plasmid and overexpressed in 293T cells for two days before harvesting for co-IP experiments using anti-HA antibody or IgG control. This is now included in the methods.

(7) Line 195-203: Authors used peaks summit distance to deduce whether TFs are binding to the same or different DNA sequences, for example they state that since YAP and TAZ ChIP-peaks summits are 25 bp apart, this indicates that YAZ/TAZ are binding the same DNA region. On the other hand, since the distance between STAT3 peaks (or JUNB peaks) and YAP/TAZ peaks is 30-44 bp, they deduce that these TFs are binding close but different DNA sequences. I am not aware of any metrics that allows these deductions, if these are based on prior knowledge, please cite the appropriate reference. Also, all these reasonings seem to be contradicted by later observations (line 235-237) where the Authors conclude that TEAD, AP1 and STAT3 TFs motifs are centered at YAP/TAZ peak summits.

We apologize for confusing language about the pairwise analysis of peak summits. We meant to say that the binding locations of all factors are extremely close together and likely coincide at the relevant motif. We did not unambiguously claim identical binding locations because the distances aren’t exactly the same and there is some error. Also, there may be some subtle effects on exactly where crosslinking occurs. For example, in previous work, we showed the NFYA and NFYB subunits of the heteromeric NFY factor have peak summits 15 bp apart and asymmetric with respect to the CCATT motif (Fleming et al., 2013).

(8) Figure 4B. color scale is missing. Also, what are the signals reported in the heatmaps? ChIP-seq signals for YAP and TAZ should be clearly indicated.(9) Figure 4C. The description of this figure is cryptic, it is unclear what TF and KO-confirmed means. As it is, it reads as if out of the 495 TAZ specific peaks, 426 are still detected in TAZ-KO (in EtOH cells). I think the Authors mean that 426 peaks are lost in the TAZ-KO.(10) Figure 5 A. Is this the intersection of ChIP-seq peaks? Please specify in the legend.

Clarified

(11) Figure 6, suppl3: it is surprising the high number of YAP or TAZ peaks, not overlapping TEAD, this is not in line with previous observations. I ask the Authors to comment on this point. Are this really TEAD independent binding or a result of poor performance of the TEAD ChIP-seq?

The original Figure 6, supplement 3 referred to motifs, not binding. This whole issue is related to point 2 and the major change in the paper.

(12) YAP/TAZ/STAT3/JUNB signatures are predictive of poor survival in TNBCs, what are these gene? Are they enriched in any particular pathway or process?

The various genes used in the survival analyses are presented in Table S3.